# Training neural operators to preserve invariant measures of chaotic attractors

**Ruoxi Jiang**[*]
Department of Computer Science
University of Chicago
Chicago, IL 60637
roxie62@uchicago.edu

**Peter Y. Lu**[*]
Department of Physics
University of Chicago
Chicago, IL 60637
lup@uchicago.edu

**Elena Orlova**
Department of Computer Science
University of Chicago
Chicago, IL 60637
eorlova@uchicago.edu

**Rebecca Willett**
Department of Statistics and Computer Science
University of Chicago
Chicago, IL 60637
willett@uchicago.edu

## Abstract

Chaotic systems make long-horizon forecasts difficult because small perturbations in initial conditions cause trajectories to diverge at an exponential rate. In this setting, neural operators trained to minimize squared error losses, while capable of accurate short-term forecasts, often fail to reproduce statistical or structural properties of the dynamics over longer time horizons and can yield degenerate results. In this paper, we propose an alternative framework designed to preserve invariant measures of chaotic attractors that characterize the time-invariant statistical properties of the dynamics. Specifically, in the multi-environment setting (where each sample trajectory is governed by slightly different dynamics), we consider two novel approaches to training with noisy data. First, we propose a loss based on the optimal transport distance between the observed dynamics and the neural operator outputs. This approach requires expert knowledge of the underlying physics to determine what statistical features should be included in the optimal transport loss. Second, we show that a contrastive learning framework, which does not require any specialized prior knowledge, can preserve statistical properties of the dynamics nearly as well as the optimal transport approach. On a variety of chaotic systems, our method is shown empirically to preserve invariant measures of chaotic attractors.

## 1   Introduction

Training fast and accurate surrogate models to emulate complex dynamical systems is key to many scientific applications of machine learning, including climate modeling [1], fluid dynamics [2, 3], plasma physics [4], and molecular dynamics [5, 6]. Fast emulators are powerful tools that can be used for forecasting and data assimilation [1, 4, 7], sampling to quantify uncertainty and compute statistical properties [1, 5, 6], identifying latent dynamical parameters [8, 9], and solving a wide range of inverse problems [10]. Specifically, neural operator architectures [2, 11, 12] have been shown to be promising physics-informed surrogate models for emulating spatiotemporal dynamics, such as dynamics governed by partial differential equations (PDEs).

---

[*]Equal contribution.

37th Conference on Neural Information Processing Systems (NeurIPS 2023).

One key feature of many of the dynamical systems in these applications is chaos, which is characterized by a high sensitivity to initial conditions and results in a theoretical limit on the accuracy of forecasts. Chaotic dynamics ensure that no matter how similarly initialized, any two distinct trajectories will diverge at an exponential rate while remaining confined to a chaotic attractor [13]. At the same time, chaos is fundamental to many critical physical processes, such as turbulence in fluid flows [14] as well as mixing and ergodicity [13]—properties that underpin the fundamental assumptions of statistical mechanics [15].

Chaos not only presents a barrier to accurate forecasts but also makes it challenging to train emulators, such as neural operators, using the traditional approach of rolling-out multiple time steps and fitting the root mean squared error (RMSE) of the prediction, as demonstrated in Fig. 1. This is because RMSE training relies on encouraging the emulator to produce more and more accurate forecasts, ideally over a long time horizon, which is severely limited by the chaotic dynamics. This problem is exacerbated by measurement noise, which, in combination with the high sensitivity to initial conditions, further degrades the theoretical limit on forecasting. Unfortunately, this is precisely the setting for many real-world scientific and engineering applications. While accurate long-term forecasts are impossible in this setting, it is still possible to replicate the statistical properties of chaotic dynamics.

Specifically, we can define a natural invariant measure on a chaotic attractor that characterizes the time-invariant statistical properties of the dynamics on the attractor [13]. By training a neural operator to preserve this invariant measure—or equivalently, preserve time-invariant statistics—we can ensure that the neural operator is properly emulating the chaotic dynamics even though it is not able to perform accurate long-term forecasts. In this paper, we introduce two new training paradigms to address this challenge. The first uses an optimal transport-based objective to ensure the invariant measure is preserved. While effective, this approach requires expert knowledge of invariant measures to determine appropriate training losses. The second paradigm uses a contrastive feature loss that naturally preserves invariant measures without this prior expert knowledge. These new losses, which are both intended to preserve the invariant statistics that govern the long-term behavior of the dynamics, are used in combination with the standard RMSE loss evaluated over a short time horizon, which ensures any remaining predictability in the short-term dynamics is correctly captured.

We operate in the multi-environment setting, in which parameters governing the system evolution may be different for each train and test sample. This setting is more challenging than the more typical single-environment setting because it requires emulators to generalize over a broader range domain. Most practical use cases for emulators—where the computational costs of generating training data and training an emulator are outweighed by the computational gains at deployment—are in the multi-environment setting.

## 1.1 Contributions

This paper makes the following key contributions. Firstly, we identify, frame, and empirically illustrate the problem of training standard neural operators on chaotic dynamics using only RMSE—namely, that the sensitivity to initial conditions in combination with noise means that any predictive model will quickly and exponentially diverge from the true trajectory in terms of RMSE. As such, RMSE is a poor signal for training a neural operator. We instead suggest training neural operators to preserve the invariant measures of chaotic attractors and the corresponding time-invariant statistics, which are robust to the combination of noise and chaos. Secondly, we propose a direct optimal transport-based approach to train neural operators to preserve the distribution of a chosen set of summary statistics. Specifically, we use a Sinkhorn divergence loss based on the Wasserstein distance to match the distribution of the summary statistics between the model predictions and the data. Next, we propose a general-purpose contrastive learning approach to learn a set of invariant statistics directly from the data without expert knowledge. Then, we construct a loss function to train neural operators to preserve these learned invariant statistics. Finally, we empirically test both of these approaches and show that the trained neural operators capture the true invariant statistics—and therefore the underlying invariant measures of the chaotic attractors—much more accurately than baseline neural operators trained using only RMSE, resulting in more stable and physically relevant long-term predictions.

## 1.2 Related work

**Neural operators.** The goal of a neural operator, proposed in the context of dynamical modeling, is to approximate the semigroup relationship between the input and output function space [2, 11, 12, 16–19]. It is particularly effective in handling complex systems governed by partial differential equations (PDEs), where the neural operator is designed to operate on an entire function or signal. The architecture designs vary significantly in recent works, including the Fourier neural operator (FNO) which uses Fourier space convolution [2], the deep operator network (DeepONet) Lu et al. [11], which consists of two subnetworks modeling the input sensors and output locations, and other modern designs like transformers [12] and graph neural networks [19].

**Multi-environment learning.** Recent developments in multi-environment learning often involve adjusting the backbone of the neural operator. For instance, Dyad [9] employs an encoder to extract time-invariant hidden features from dynamical systems, utilizing a supervisory regression loss relative to the context values. These hidden features are then supplied as additional inputs to the neural operator for decoding in a new environment. In contrast, Context-informed Dynamics Adaptation (CoDA, Kirchmeyer et al. [20]) decodes the environment context via a hypernetwork, integrating the parameters of the hypernetwork as an auxiliary input to the neural operator. Like our approach, Dyad learns time-invariant features from dynamical systems. However, their methods require a time series of measurements instead of just initial conditions. In addition, both CoDA and Dyad alter the backbone of the neural operator, making it challenging to adapt to new neural operator architectures. In contrast, our method uses the learned time-invariant features to determine a loss function that can be applied to any neural operator architecture without changes to either the backbone or the inputs.

**Long-term predictions.** Lu et al. [21] try to recover long-term statistics by learning a discrete-time stochastic reduced-order system. From the perspective of training objectives, Li et al. [16] propose using a Sobolev norm and dissipative regularization to facilitate learning of long-term invariant measures. The Sobolev norm, which depends on higher-order derivatives of both the true process and the output of the neural operator, can capture high-frequency information and is superior to using only RMSE. However, its effectiveness diminishes in noisy and chaotic scenarios where it struggles to capture the correct statistics. The dissipative loss term, regularizes the movement of the neural operator with respect to the input, and attempts to make the emulator stay on the attractor. However, in the case of a multi-environment setup for chaotic systems, our goal is to learn a neural operator capable of modeling long-term distributions with regard to different contexts. Using the dissipative loss as a regularization will cause the neural operator to be insensitive to the context and fail to model different attractors.

Prior [22] and concurrent work [23] on training recurrent neural networks (RNNs) have suggested that using teacher forcing methods with a squared error loss on short time sequences can produce high-quality emulators for chaotic time series. Instead, we focus on training neural operators on fully observed high-dimensional deterministic chaos and find that using only an RMSE loss generally fails to perform well on noisy chaotic dynamics. Other concurrent work [24] has suggested using dynamical invariants such as the Lyapunov spectrum and the fractal dimension to regularize training for reservoir computing emulators. We show that emulators trained using our approaches also correctly capture the Lyapunov spectrum and fractal dimension of the chaotic attractors without explicitly including these invariants, which are difficult to estimate in high dimensions, in our loss.

**Wasserstein distance and optimal transport.** Optimal transport theory and the Wasserstein metric have become powerful theoretical and computational tools. for a variety of applications, including generative computer vision models [25], geometric machine learning and data analysis [26], particle physics [27], as well as identify conservation laws [28] and fitting parameterized models for low-dimensional dynamical systems [29]. In particular, Yang et al. [29] uses the Wasserstein distance as an optimization objective to directly fit a parameterized model to data from a low-dimensional attractor. They compute the Wasserstein distance by turning the nonlinear dynamics in the original state space into a linear PDE in the space of distributions and then solving a PDE-constrained optimization problem. This has some nice theoretical properties but generally scales poorly to high-dimensional dynamical systems. Similarly, concurrent work [30] suggests modeling dynamics by fitting a Fokker–Planck PDE for the probability density of the state using the Wasserstein distance. However, since this also requires estimating the full probability distribution of the state on a mesh grid, it is again challenging to scale this method to high-dimensional systems. In contrast, our proposed optimal transport approach focuses on fitting deep learning-based neural operators to high-dimensional chaotic

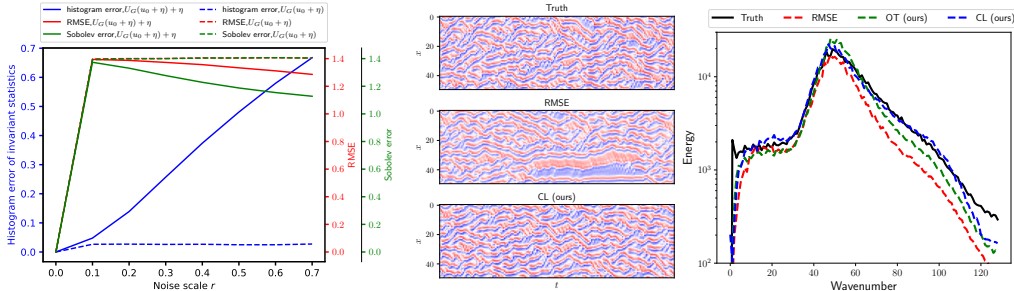

(a) Error metrics vs. noise scale $r$.    (b) Predictions from emulators.    (c) Energy spectrum.

Figure 1: **The impact of noise on invariant statistics vs. RMSE and Sobolev norm.** (a) We show the impact of noise on various error metrics using ground truth simulations of the chaotic Kuramoto–Sivashinsky (KS) system with increasingly noisy initial conditions $\mathbf{U}_G(\mathbf{u}_0 + \eta)$ as well as with added measurement noise $\mathbf{U}_G(\mathbf{u}_0 + \eta) + \eta$. Here, $\mathbf{U}_G(\cdot)$ refers to the ground truth solution to the differential equation (1) for the KS system given an initial condition, and $\eta \sim \mathcal{N}(0, r^2\sigma^2 I)$, where $\sigma^2$ is the temporal variance of the trajectory $\mathbf{U}_G(\mathbf{u}_0)$ and $r$ is the noise scale. Relative RMSE and Sobolev norm [16], which focus on short-term forecasts, deteriorate rapidly with noise $\eta$, whereas the invariant statistics have a much more gradual response to noise, indicating robustness. (b) The emulator trained with only RMSE degenerates at times into striped patches, while ours is much more statistically consistent with the ground truth. (c) Again, the emulator trained with only RMSE performs the worst in terms of capturing the expected energy spectrum over a long-term prediction.

dynamical systems by first choosing a set of physically relevant summary statistics and then using the computationally efficient Sinkhorn algorithm [31] to compute our optimal transport loss—an entropy-regularized approximation for the (squared) Wasserstein distance—on the distribution of the summary statistics.

**Feature loss and contrastive learning.** Contrastive learning [32–34] charges an encoder with generating features that are unaffected by transformations by promoting the similarity of features between different transformations of the same image and by maximizing the feature distance between distinct images. The recent advancements in contrastive learning mainly result from using extensive data augmentation to motivate the encoder to learn semantic representations. In addition to its success in image recognition, contrastive learning has proven to be effective in modeling fine-grained structures [35] and in scientific applications [10]. Training data generators or solving inverse problems generally requires output to be coherent across multiple structural measurements, a requirement that pixel-wise MSE typically fails to meet. For this reason, Johnson et al. [36] computes MSE on deep features of the classification models as a structural distance metric to train generative models. Tian et al. [37] propose the encoder of contrastive learning as an unsupervised alternative for calculating feature loss.

## 2   Problem Formulation

Consider a dynamical system with state space $\mathbf{u} \in \mathcal{U}$ governed by

$$\frac{\mathrm{d}\mathbf{u}}{\mathrm{d}t} = G(\mathbf{u}, \phi), \tag{1}$$

where $G$ is the governing function, and $\phi \in \Phi$ is a set of parameters that specify the dynamics. We will denote trajectories governed by these dynamics (1) as $\mathbf{u}(t)$, particular points on the trajectory as $\mathbf{u}_t \in \mathcal{U}$, and a sequence of $K + 1$ consecutive time points on the trajectory as $\mathbf{U}_{I:I+K} := \{\mathbf{u}_{t_i}\}_{i=I}^{I+K}$. Our experiments use equally spaced time points with a time step $\Delta t$. We refer to settings where $\phi$ varies as *multi-environment settings*.

**Our goal** is to learn an emulator in a multi-environment setting that approximates the dynamics (1) with a data-driven neural operator $\hat{g}_\theta : \mathcal{U} \times \Phi \to \mathcal{U}$ that makes discrete predictions

$$\hat{\mathbf{u}}_{t+\Delta t} := \hat{g}_\theta(\hat{\mathbf{u}}_t, \phi), \tag{2}$$

where $\theta$ are the parameters of the neural operator $\hat{g}_\theta$. In the multi-environment setting, we will have data from a variety of environments $n \in \{1, 2, \ldots, N\}$, and thus our training data $\{\mathbf{U}_{0:L}^{(n)}, \phi^{(n)}\}_{n=1}^N$ consists of trajectories $\mathbf{U}_{0:L}^{(n)}$ and the corresponding environment parameters $\phi^{(n)}$.

**Predicted sequence notation.** We will denote a sequence of $h+1$ autonomously predicted states (using the neural operator $\hat{g}_\theta$) with an initial condition $\mathbf{u}_{t_I}^{(n)}$ as

$$\hat{\mathbf{U}}_{I:I+h}^{(n)} := \{\mathbf{u}_{t_I}^{(n)}, \hat{g}_\theta(\mathbf{u}_{t_I}^{(n)}, \phi^{(n)}), \hat{g}_\theta \circ \hat{g}_\theta(\mathbf{u}_{t_I}^{(n)}, \phi^{(n)}), \ldots, \overbrace{\hat{g}_\theta \circ \cdots \circ \hat{g}_\theta}^{h}(\mathbf{u}_{t_I}^{(n)}, \phi^{(n)})\}. \tag{3}$$

We will often use a concatenated sequence (with total length $K+1$) of these autonomous prediction sequences (each with length $h+1$), which we will denote as

$$\hat{\mathbf{U}}_{I:h:I+K}^{(n)} := \hat{\mathbf{U}}_{I:I+h}^{(n)} \oplus \hat{\mathbf{U}}_{I+h+1:I+2h+1}^{(n)} \oplus \cdots \oplus \hat{\mathbf{U}}_{I+K-h:I+K}^{(n)}, \tag{4}$$

where $\oplus$ is concatenation.

**Chaotic dynamical systems and invariant measures of chaotic attractors.** We focus on chaotic dynamical systems that have one or more chaotic attractors, which exhibit a sensitive dependence on initial conditions characterized by a positive Lyapunov exponent [13]. For each chaotic attractor $\mathcal{A}$, we can construct a natural invariant measure (also known as a physical measure)

$$\mu_\mathcal{A} = \lim_{T \to \infty} \frac{1}{T} \int^T \delta_{\mathbf{u}_\mathcal{A}(t)} \, \mathrm{d}t, \tag{5}$$

where $\delta_{\mathbf{u}(t)}$ is the Dirac measure centered on a trajectory $\mathbf{u}_\mathcal{A}(t)$ that is in the basin of attraction of $\mathcal{A}$ [13]. Note that, because $\mathcal{A}$ is an attractor, any trajectory $\mathbf{u}_\mathcal{A}(t)$ in the basin of attraction of $\mathcal{A}$ will give the same invariant measure $\mu_\mathcal{A}$, i.e. the dynamics are ergodic on $\mathcal{A}$. Therefore, any time-invariant statistical property $S_\mathcal{A}$ of the dynamics on $\mathcal{A}$ can be written as

$$S_\mathcal{A} = \mathbb{E}_{\mu_\mathcal{A}}[s] = \int s(\mathbf{u}) \, \mathrm{d}\mu_\mathcal{A}(\mathbf{u}) = \lim_{T \to \infty} \frac{1}{T} \int^T s(\mathbf{u}_\mathcal{A}(t)) \, \mathrm{d}t \tag{6}$$

for some function $s(\mathbf{u})$ and a trajectory $\mathbf{u}_\mathcal{A}(t)$ in the basin of attraction of $\mathcal{A}$. Conversely, for any $s(\mathbf{u})$, (6) gives a time-invariant property $S_\mathcal{A}$ of the dynamics.

In this work, we assume each sampled trajectory in the data is from a chaotic attractor and therefore, has time-invariant statistical properties characterized by the natural invariant measure of the attractor.

**Noisy measurements.** Because of the sensitivity to initial conditions, accurate long-term forecasts (in terms of RMSE) are not possible for time scales much larger than the Lyapunov time [13]. This is because any amount of noise or error in the measurement or forecast model will eventually result in exponentially diverging trajectories. The noisier the data, the more quickly this becomes a problem (Fig. 1). However, in the presence of measurement noise, invariant statistics of the noisy trajectory

$$\tilde{\mathbf{u}}_\mathcal{A}(t) = \mathbf{u}_\mathcal{A}(t) + \eta, \quad \eta \sim p_\eta \tag{7}$$

can still provide a useful prediction target. The invariant statistics will be characterized by a broadened measure (the original measure convolved with the noise distribution $p_\eta$)

$$\tilde{\mu}_\mathcal{A} = \lim_{T \to \infty} \frac{1}{T} \int^T \delta_{\tilde{\mathbf{u}}_\mathcal{A}(t)} \, \mathrm{d}t = \mu_\mathcal{A} * p_\eta = \lim_{T \to \infty} \frac{1}{T} \int^T p_\eta(\mathbf{u}_\mathcal{A}(t)) \, \mathrm{d}t \tag{8}$$

and are therefore given by

$$\begin{aligned}
\tilde{S}_\mathcal{A} = \mathbb{E}_{\tilde{\mu}_\mathcal{A}}[s] &= \int s(\mathbf{u}) \, \mathrm{d}\tilde{\mu}_\mathcal{A}(\mathbf{u}) \\
&= \lim_{T \to \infty} \frac{1}{T} \int^T s(\tilde{\mathbf{u}}_\mathcal{A}(t)) \, \mathrm{d}t = \lim_{T \to \infty} \frac{1}{T} \int^T s(\mathbf{u}_\mathcal{A}(t)) \, p_\eta(\mathbf{u}_\mathcal{A}(t)) \, \mathrm{d}t,
\end{aligned} \tag{9}$$

which can be a good approximation for $S_\mathcal{A} = \mathbb{E}_{\mu_\mathcal{A}}[s]$ even in noisy conditions and does not suffer from the exponential divergence of RMSE (Fig. 1).

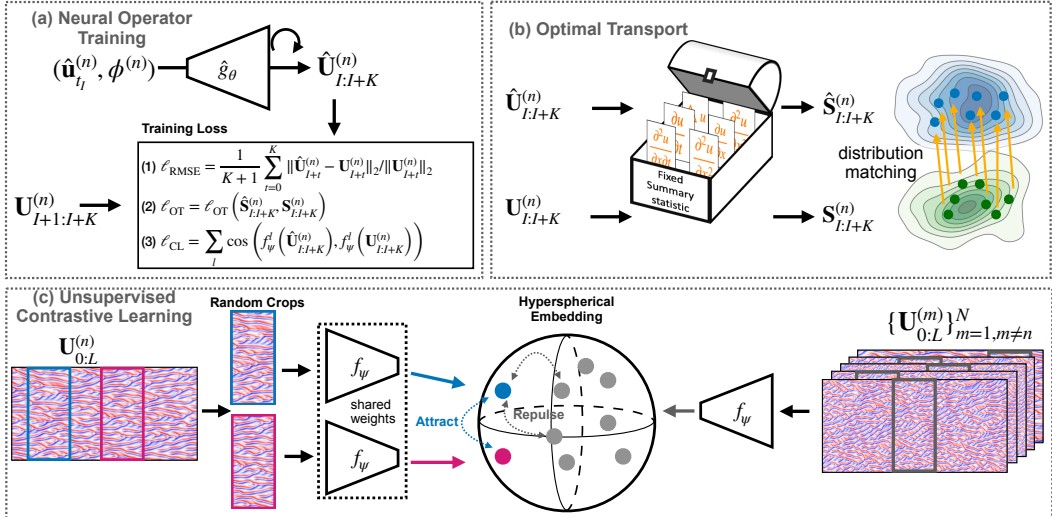

Figure 2: **Our proposed approaches for training neural operators.** (a) Neural operators are emulators trained to take an initial state and output future states in a recurrent fashion. To ensure the neural operator respects the statistical properties of chaotic dynamics when trained on noisy data, we propose two additional loss functions for matching relevant long-term statistics. (b) We match the distribution of summary statistics, chosen based on prior knowledge, between the emulator predictions and noisy data using an optimal transport loss. (c) In the absence of prior knowledge, we take advantage of self-supervised contrastive learning to automatically learn relevant time-invariant statistics, which can then be used to train neural operators.

## 3 Proposed Approaches

### 3.1 Physics-informed optimal transport

We propose an optimal transport-based loss function to match the distributions of a set of summary statistics $\mathbf{s}(\mathbf{u}) = \left[s^{(1)}(\mathbf{u}), s^{(2)}(\mathbf{u}), \ldots, s^{(k)}(\mathbf{u})\right]$ between the data $\mathbf{s}_i = \mathbf{s}(\mathbf{u}_{t_i}) \sim \mu_{\mathbf{s}}$ and the model predictions $\hat{\mathbf{s}}_j = \mathbf{s}(\hat{\mathbf{u}}_{t_j}) \sim \mu_{\hat{\mathbf{s}}}$. Here, $\mu_{\mathbf{s}}$ and $\mu_{\hat{\mathbf{s}}}$ are the probability measures of the summary statistics for the true chaotic attractor $\mathcal{A}$ and the attractor $\hat{\mathcal{A}}$ learned by the neural operator, respectively. Specifically, $\mu_{\mathbf{s}}(\mathbf{s}') = \int \delta_{\mathbf{s}(\mathbf{u})-\mathbf{s}'} \mu_{\mathcal{A}}(\mathbf{u}) \, d\mathbf{u}$ and $\mu_{\hat{\mathbf{s}}}(\hat{\mathbf{s}}') = \int \delta_{\mathbf{s}(\hat{\mathbf{u}})-\hat{\mathbf{s}}'} \mu_{\hat{\mathcal{A}}}(\hat{\mathbf{u}}) \, d\hat{\mathbf{u}}$. By choosing the set of summary statistics using expert domain knowledge, this approach allows us to directly guide the neural operator toward preserving important physical properties of the system.

**Optimal transport—Wasserstein distance.** We match the distributions of summary statistics using the Wasserstein distance $W(\mu_{\mathbf{s}}, \mu_{\hat{\mathbf{s}}})$—a distance metric between probability measures defined in terms of the solution to an optimal transport problem:

$$\frac{1}{2} W(\mu_{\mathbf{s}}, \mu_{\hat{\mathbf{s}}})^2 := \inf_{\pi \in \Pi(\mu_{\mathbf{s}}, \mu_{\hat{\mathbf{s}}})} \int c(\mathbf{s}, \hat{\mathbf{s}}) \, d\pi(\mathbf{s}, \hat{\mathbf{s}}). \tag{10}$$

We use a quadratic cost function $c(\mathbf{s}, \hat{\mathbf{s}}) = \frac{1}{2} \|\mathbf{s} - \hat{\mathbf{s}}\|^2$ (i.e., $W$ is the 2-Wasserstein distance), and the map $\pi \in \Pi(\mu_{\mathbf{s}}, \mu_{\hat{\mathbf{s}}})$ must be a valid transport map between $\mu_{\mathbf{s}}$ and $\mu_{\hat{\mathbf{s}}}$ (i.e., a joint distribution for $\mathbf{s}, \hat{\mathbf{s}}$ with marginals that match $\mu_{\mathbf{s}}$ and $\mu_{\hat{\mathbf{s}}}$) [38].

In the discrete setting with distributions represented by samples $\mathbf{S} = \{\mathbf{s}_i\}_{i=1}^L$ and $\hat{\mathbf{S}} = \{\hat{\mathbf{s}}_j\}_{j=1}^L$, the Wasserstein distance is given by

$$\frac{1}{2} W(\mathbf{S}, \hat{\mathbf{S}})^2 := \min_{T \in \Pi} \sum_{i,j} T_{ij} C_{ij}, \tag{11}$$

where the cost matrix $C_{ij} = \frac{1}{2} \|\mathbf{s}_i - \hat{\mathbf{s}}_j\|^2$. The set of valid discrete transport maps $\Pi$ consists of all matrices $T$ such that $\forall i, j, T_{ij} \geq 0$, $\sum_j T_{ij} = 1$, and $\sum_i T_{ij} = 1$.

**Entropy-regularized optimal transport—Sinkhorn divergence.** Exactly solving the optimal transport problem associated with computing the Wasserstein distance is computationally prohibitive,

especially in higher dimensions. A common approximation made to speed up computation is to introduce an entropy regularization term, resulting in a convex relaxation of the original optimal transport problem:

$$\frac{1}{2} W^\gamma(\mathbf{S}, \hat{\mathbf{S}})^2 := \min_{T \in \Pi} \sum_{i,j} T_{ij} C_{ij} - \gamma\, h(T), \tag{12}$$

where $h(T) = -\sum_{i,j} T_{ij} \log T_{ij}$ is the entropy of the transport map. This entropy-regularized optimal transport problem can be solved efficiently using the Sinkhorn algorithm [31].

As $\gamma \to 0$, the entropy-regularized Wasserstein distance $W^\gamma \to W^0 = W$ reduces to the exact Wasserstein distance (11). For $\gamma > 0$, we can further correct for an entropic bias to obtain the Sinkhorn divergence [39, 40]

$$\ell_{\mathrm{OT}}(\mathbf{S}, \hat{\mathbf{S}}) = \frac{1}{2} \overline{W}^\gamma(\mathbf{S}, \hat{\mathbf{S}})^2 := \frac{1}{2} \left( W^\gamma(\mathbf{S}, \hat{\mathbf{S}})^2 - \frac{W^\gamma(\mathbf{S}, \mathbf{S})^2 + W^\gamma(\hat{\mathbf{S}}, \hat{\mathbf{S}})^2}{2} \right), \tag{13}$$

which gives us our optimal transport loss. Combined with relative root mean squared error (RMSE)

$$\ell_{\mathrm{RMSE}}(\mathbf{U}, \hat{\mathbf{U}}) := \frac{1}{K+1} \sum_{\mathbf{u}_t, \hat{\mathbf{u}}_t \in \mathbf{U}, \hat{\mathbf{U}}} \frac{\|\mathbf{u}_t - \hat{\mathbf{u}}_t\|_2}{\|\mathbf{u}_t\|_2} \tag{14}$$

for short-term prediction consistency [2, 16, 18], our final loss function is

$$\ell(\theta) = \mathop{\mathbb{E}}_{\substack{n \in \{1,\dots,N\} \\ I \in \{0,\dots,L-K\}}} \left[ \alpha\, \ell_{\mathrm{OT}}\big(\mathbf{S}_{I:I+K}^{(n)}, \hat{\mathbf{S}}_{I:h:I+K}^{(n)}\big) + \ell_{\mathrm{RMSE}}\big(\mathbf{U}_{I:I+K}^{(n)}, \hat{\mathbf{U}}_{I:h_{\mathrm{RMSE}}:I+K}^{(n)}\big) \right], \tag{15}$$

where $\mathbf{S}_{I:I+K}^{(n)} := \big\{ \mathbf{s}(\mathbf{u}) \mid \mathbf{u} \in \mathbf{U}_{I,I+K}^{(n)} \big\}$, $\hat{\mathbf{S}}_{I:h:I+K}^{(n)} := \big\{ \mathbf{s}(\hat{\mathbf{u}}) \mid \hat{\mathbf{u}} \in \hat{\mathbf{U}}_{I:h:I+K}^{(n)} \big\}$, and $\alpha > 0$ is a hyperparameter. Note that $\hat{\mathbf{S}}_{I:h:I+K}^{(n)}$ and $\hat{\mathbf{U}}_{I:h_{\mathrm{RMSE}}:I+K}^{(n)}$ implicitly depend on weights $\theta$.

## 3.2 Contrastive feature learning

When there is an absence of prior knowledge pertaining to the underlying dynamical system, or when the statistical attributes are not easily differentiable, we propose an alternative contrastive learning-based approach to learn the relevant invariant statistics directly from the data. We first use contrastive learning to train an encoder to capture invariant statistics of the dynamics in the multi-environment setting. We then leverage the feature map derived from this encoder to construct a feature loss that preserves the learned invariant statistics during neural operator training.

**Contrastive learning.** The objective of self-supervised learning is to train an encoder $f_\psi(\mathbf{U})$ (with parameters $\psi$) to compute relevant invariant statistics of the dynamics from sequences $\mathbf{U}$ with fixed length $K + 1$. We do not explicitly train on the environment parameters $\phi$ but rather use a general-purpose contrastive learning approach that encourages the encoder $f_\psi$ to learn a variety of time-invariant features that are able to distinguish between sequences from different trajectories (and therefore different $\phi$).

A contrastive learning framework using the Noise Contrastive Estimation (InfoNCE) loss has been shown to preserve context-aware information by training to match sets of positive pairs while treating all other combinations as negative pairs [32]. The selection of positive pairs is pivotal to the success of contrastive learning. In our approach, the key premise is that two sequences $\mathbf{U}_{I:I+K}^{(n)}$, $\mathbf{U}_{J:J+K}^{(n)}$ from the same trajectory $\mathbf{U}_{0:L}^{(n)}$ both sample the same chaotic attractor, i.e. their statistics should be similar, so we treat any such pair of sequences as positive pairs. Two sequences $\mathbf{U}_{I:I+K}^{(n)}$, $\mathbf{U}_{H:H+K}^{(m)}$ from different trajectories are treated as negative pairs. This allows us to formulate the InfoNCE loss as:

$$\ell_{\mathrm{InfoNCE}}(\psi; \tau) :=$$

$$\mathop{\mathbb{E}}_{\substack{n \in \{1,\dots,N\} \\ I,J \in \{0,\dots,L-K\}}} \left[ -\log \left( \frac{\exp\Big( \langle f_\psi(\mathbf{U}_{I:I+K}^{(n)}), f_\psi(\mathbf{U}_{J:J+K}^{(n)}) \rangle / \tau \Big)}{\mathop{\mathbb{E}}_{\substack{m \neq n \\ H \in \{0,\dots,L-K\}}} \Big[ \exp\Big( \langle f_\psi(\mathbf{U}_{I:I+K}^{(n)}), f_\psi(\mathbf{U}_{H:H+K}^{(m)}) \rangle / \tau \Big) \Big]} \right) \right]. \tag{16}$$

The term in the numerator enforces alignment of the positive pairs which ensures we obtain time-invariant statistics, while the term in the numerator encourages uniformity, i.e. maximizing mutual information between the data and the embedding, which ensures we can distinguish between negative pairs from different trajectories [41]. This provides intuition for why our learned encoder $f_\psi$ identifies relevant time-invariant statistics that can distinguish different chaotic attractors.

**Contrastive feature loss.** To construct our feature loss, we use the cosine distance between a series of features of the encoder network $f_\psi$ [42]:

$$\ell_{\mathrm{CL}}\big(\mathbf{U}, \hat{\mathbf{U}}; f_\psi\big) := \sum_l \cos\big(f_\psi^l(\mathbf{U}), f_\psi^l(\hat{\mathbf{U}})\big), \tag{17}$$

where $f_\psi^l$ gives the output the $l$-th layer of the neural network. The combined loss that we use for training the neural operator is given by

$$\ell(\theta) = \mathop{\mathbb{E}}_{\substack{n \in \{1,\ldots,N\} \\ I \in \{0,\ldots,L-K\}}} \left[ \lambda\, \ell_{\mathrm{CL}}\big(\mathbf{U}_{I:I+K}^{(n)}, \hat{\mathbf{U}}_{I:h:I+K}^{(n)}; f_\psi\big) + \ell_{\mathrm{RMSE}}\big(\mathbf{U}_{I:I+K}^{(n)}, \hat{\mathbf{U}}_{I:h_{\mathrm{RMSE}}:I+K}^{(n)}\big) \right], \tag{18}$$

where $\lambda > 0$ is a hyperparameter.

## 4 Experiments

We evaluate our approach on the 1D chaotic Kuramoto–Sivanshinsky (KS) system and a finite-dimensional Lorenz 96 system. In all cases, we ensure that the systems under investigation remain in chaotic regimes. We demonstrate the effectiveness of our approach in preserving key statistics in these unpredictable systems, showcasing our ability to handle the complex nature of chaotic systems. The code is available at: `https://github.com/roxie62/neural_operators_for_chaos`.

**Experimental setup.** Our data consists of noisy observations $\mathbf{u}(t)$ with noise $\eta \sim \mathcal{N}(0, r^2\sigma^2 I)$, where $\sigma^2$ is the temporal variance of the trajectory and $r$ is a scaling factor. **Baselines.** We primarily consider the baseline as training with RMSE [2]. We have additional baselines in Appendix B, including Gaussian denoising and a Sobolev norm loss with dissipative regularization [16]. **Backbones.** We use the Fourier neural operator (FNO, [18]). **Evaluation metrics.** We use a variety of statistics-based evaluation metrics and other measures that characterize the chaotic attractor. See Appendix C.1 for details.

### 4.1 Lorenz-96

As is a common test model for climate systems, data assimilation, and other geophysical applications [43–45], the Lorenz-96 system is a key tool for studying chaos theory, turbulence, and nonlinear dynamical systems. It is described by the differential equation

$$\frac{du_i}{dt} = (u_{i+1} - u_{i-2})u_{i-1} - u_i + F \tag{19}$$

Its dynamics exhibit strong energy-conserving non-linearity, and for a large $F \geq 10$, it can exhibit strong chaotic turbulence and symbolizes the inherent unpredictability of the Earth's climate.

**Experimental setup.** When using optimal transport loss, we assume that expert knowledge is derived from the underlying equation. For Lorenz-96, we define the relevant statistics as $s(\mathbf{u}) := \{\frac{du_i}{dt}, (u_{i+1} - u_{i-2})u_{i-1}, u_i\}$. We generate 2000 training data points with each $\phi^{(n)}$ randomly sampled from a uniform distribution with the range $[10.0, 18.0]$. We vary the noise level $r$ from 0.1 to 0.3 and show consistent improvement in the relevant statistics. **Results.** The results are presented in Table 1, and predictions and invariant statistics are shown in Fig. 3 (refer to C.4 for more visualizations).

### 4.2 Kuramoto–Sivashinsky

Known as a model for spatiotemporal chaos, Kuramoto–Sivashinsky (KS) has been widely used to describe various physical phenomena, including fluid flows in pipes, plasma physics, and dynamics of certain chemical reactions [46]. It captures wave steepening via the nonlinear term $u\frac{\partial u}{\partial x}$, models dispersion effects through $\frac{\partial^2 u}{\partial x^2}$, and manages discontinuities by introducing hyper-viscosity via $\frac{\partial^4 u}{\partial x^4}$:

$$\frac{\partial u}{\partial t} = -u\frac{\partial u}{\partial x} - \phi\frac{\partial^2 u}{\partial x^2} - \frac{\partial^4 u}{\partial x^4}. \tag{20}$$

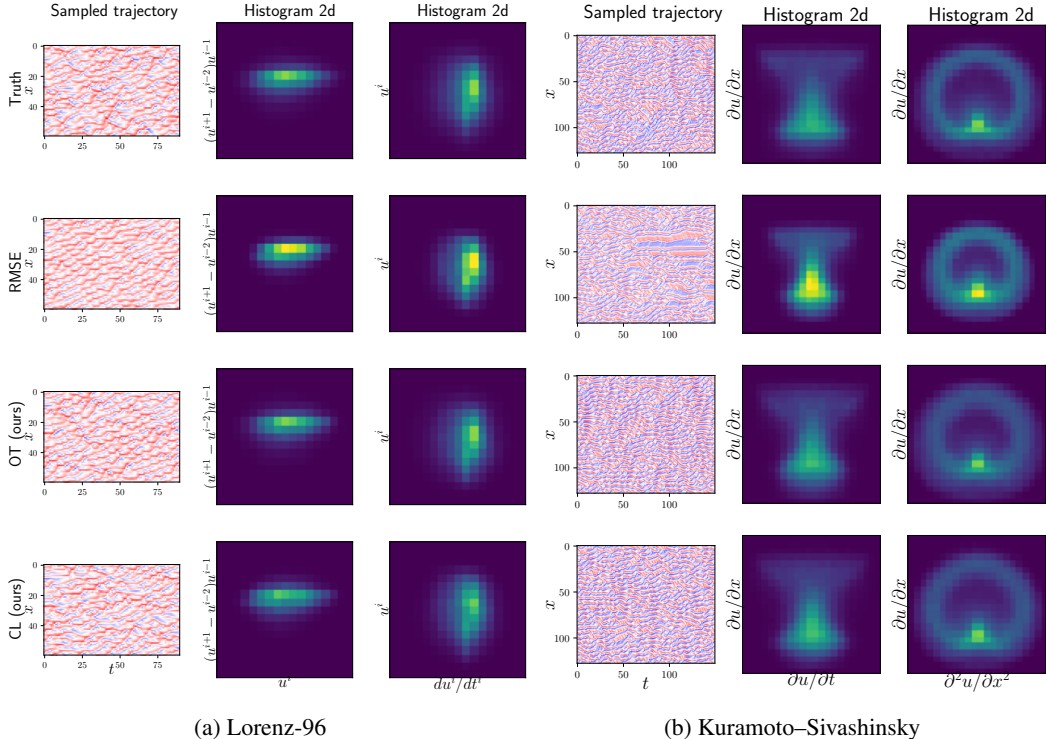

|       | (a) Lorenz-96 | (b) Kuramoto–Sivashinsky |

Figure 3: **Sampled emulator dynamics and summary statistic distributions.** We evaluate our proposed approaches by comparing them to a baseline model that is trained solely using relative RMSE loss. We conduct this comparison on two dynamical systems: (a) Lorenz-96 and (b) Kuramoto–Sivashinsky (KS). For each system, we show a visual comparison of the predicted dynamics (left) and two-dimensional histograms of relevant statistics (middle and right). We observe that training the neural operator with our proposed optimal transport (OT) or contrastive learning (CL) loss significantly enhances the long-term statistical properties of the emulator, as seen in the raw emulator dynamics and summary statistic distributions. The performance of the CL loss, which uses no prior knowledge, is comparable to that of the OT loss, which requires an explicit choice of summary statistics.

**Experimental setup.** For the KS system, we define $\mathbf{s}(\mathbf{u}) := \left\{ \frac{\partial u}{\partial t}, \frac{\partial u}{\partial x}, \frac{\partial^2 u}{\partial x^2} \right\}$ [47]. We generate 2000 training data points, with each $\phi^{(n)}$ being randomly selected from a uniform distribution within the range of $[1.0, 2.6]$. **Results.** We report our results over 200 test instances in Table 2 and visualize the predictions and invariant statistics in Fig. 3.

## 5 Discussion and Limitations

We have demonstrated two approaches for training neural operators on chaotic dynamics by preserving the invariant measures of the chaotic attractors. The optimal transport approach uses knowledge of the underlying system to construct a physics-informed objective that matches relevant summary statistics, while the contrastive learning approach uses a multi-environment setting to directly learn relevant invariant statistics. In both cases, we see a significant improvement in the ability of the emulator to reproduce the invariant statistics of chaotic attractors from noisy data when compared with traditional RMSE-only training that focuses on short-term forecasting.

We also find, in high-noise settings, that both of our approaches give similar or lower leading LE errors than an emulator trained on RMSE loss alone, despite the fact that our method is only encouraged to match invariant statistics of the final attractor rather than dynamical quantities. We also see evidence that the fractal dimension of the contrastive learning approach is closer to the true attractor. However, we note that the fractal dimension is difficult to reliably estimate for high-dimensional chaotic attractors [48].

| $r$ | Training | Histogram Error ↓ | Energy Spec. Error ↓ | Leading LE Error ↓ | FD Error ↓ |
|---|---|---|---|---|---|
| | $\ell_{\mathrm{RMSE}}$ | 0.056 (0.051, 0.062) | 0.083 (0.078, 0.090) | **0.013** (0.006, 0.021) | 1.566 (0.797, 2.309) |
| 0.1 | $\ell_{\mathrm{OT}} + \ell_{\mathrm{RMSE}}$ | **0.029** (0.027, 0.032) | **0.058** (0.052, 0.064) | 0.050 (0.040, 0.059) | 1.424 (0.646, 2.315) |
| | $\ell_{\mathrm{CL}} + \ell_{\mathrm{RMSE}}$ | 0.033 (0.029, 0.037) | **0.058** (0.049, 0.065) | 0.065 (0.058, 0.073) | **1.042** (0.522, 1.685) |
| | $\ell_{\mathrm{RMSE}}$ | 0.130 (0.118, 0.142) | 0.182 (0.172, 0.188) | 0.170 (0.156, 0.191) | 2.481 (1.428, 3.807) |
| 0.2 | $\ell_{\mathrm{OT}} + \ell_{\mathrm{RMSE}}$ | **0.039** (0.035, 0.042) | **0.086** (0.079, 0.095) | 0.016 (0.006, 0.030) | 2.403 (1.433, 3.768) |
| | $\ell_{\mathrm{CL}} + \ell_{\mathrm{RMSE}}$ | 0.073 (0.066, 0.080) | 0.131 (0.117, 0.149) | **0.012** (0.006, 0.018) | **1.681** (0.656, 2.682) |
| | $\ell_{\mathrm{RMSE}}$ | 0.215 (0.204, 0.234) | 0.291 (0.280, 0.305) | 0.440 (0.425, 0.463) | 3.580 (2.333, 4.866) |
| 0.3 | $\ell_{\mathrm{OT}} + \ell_{\mathrm{RMSE}}$ | **0.057** (0.052, 0.064) | **0.123** (0.116, 0.135) | 0.084 (0.062, 0.134) | 3.453 (2.457, 4.782) |
| | $\ell_{\mathrm{CL}} + \ell_{\mathrm{RMSE}}$ | 0.132 (0.111, 0.151) | 0.241 (0.208, 0.285) | **0.064** (0.045, 0.091) | **1.894** (0.942, 3.108) |

Table 1: **Emulator performance on Lorenz-96 data with varying noise scale** $r = 0.1, 0.2, 0.3$**.** The median (25th, 75th percentile) of the evaluation metrics (Appendix C.1) are computed on 200 Lorenz-96 test instances (each with 1500 time steps) for the neural operator trained with (1) only RMSE loss $\ell_{\mathrm{RMSE}}$; (2) optimal transport (OT) and RMSE loss $\ell_{\mathrm{OT}} + \ell_{\mathrm{RMSE}}$ (using prior knowledge to choose summary statistics); and (3) contrastive learning (CL) and RMSE loss $\ell_{\mathrm{CL}} + \ell_{\mathrm{RMSE}}$ (without prior knowledge). We show significant improvements on the long-term statistical metrics including $L_1$ histogram error of the chosen statistics $\mathbf{S}(\mathbf{u}) := \{ \frac{du_i}{dt}, (u_{i+1} - u_{i-2})u_{i-1}, u_i \}$; relative error of Fourier energy spectrum; and absolute error of estimated fractal dimension (FD). For high noise, OT and CL training also improve the leading Lyapunov exponent (LE) of the neural operator.

| Training | Histogram Error ↓ | Energy Spec. Error ↓ | Leading LE Error ↓ |
|---|---|---|---|
| $\ell_{\mathrm{RMSE}}$ | 0.390 (0.325, 0.556) | 0.290 (0.225, 0.402) | 0.101 (0.069, 0.122) |
| $\ell_{\mathrm{OT}} + \ell_{\mathrm{RMSE}}$ | **0.172** (0.146, 0.197) | 0.211 (0.188, 0.250) | **0.094** (0.041, 0.127) |
| $\ell_{\mathrm{CL}} + \ell_{\mathrm{RMSE}}$ | 0.193 (0.148, 0.247) | **0.176** (0.130, 0.245) | 0.108 (0.068, 0.132) |

Table 2: **Emulator performance on Kuramoto–Sivashinsky data with noise scale** $r = 0.3$**.** The median (25th, 75th percentile) of the evaluation metrics (Appendix C.1) are computed on 200 Kuramoto–Sivashinsky test instances (each with 1000 time steps) for the neural operator trained with (1) only RMSE loss $\ell_{\mathrm{RMSE}}$; (2) optimal transport (OT) and RMSE loss $\ell_{\mathrm{OT}} + \ell_{\mathrm{RMSE}}$ (using prior knowledge to choose summary statistics); and (3) contrastive learning (CL) and RMSE loss $\ell_{\mathrm{CL}} + \ell_{\mathrm{RMSE}}$ (without prior knowledge). We again show significant improvements in the long-term statistical metrics including $L_1$ histogram error of the chosen statistics $\mathbf{S}(\mathbf{u}) := \{ \frac{\partial u}{\partial t}, \frac{\partial u}{\partial x}, \frac{\partial^2 u}{\partial x^2} \}$ and relative error of Fourier energy spectrum. The fractal dimension (FD) is highly unstable in high dimensions [48] and could not be estimated for this dataset.

**Limitations.** Because we rely on invariant measures, our current approach is limited to trajectory data from attractors, i.e. we assume that the dynamics have reached an attractor and are not in a transient phase. We also cannot handle explicit time dependence, including time-dependent forcing or control parameters. For the optimal transport approach, choosing informative summary statistics based on prior knowledge is key to good performance (Appendix B.3). For the contrastive learning approach, the quality of the learned invariant statistics also depends on the diversity of the environments present in the multi-environment setting, although our additional experiments show that we can still obtain good performance even with minimal environment diversity (Appendix B.4).

**Future work.** In the future, we may be able to adapt our approaches to allow for mild time dependence by restricting the time range over which we compute statistics and select positive pairs. This would allow us to study slowly varying dynamics as well as sharp discrete transitions such as tipping points. We can also improve the diversity of the data for contrastive learning by designing new augmentations or using the training trajectory of the neural operator to generate more diverse negative pairs. We will investigate generalizing our approaches to other difficult systems, such as stochastic differential equations or stochastic PDEs, and we would like to further study the trade-offs and synergies between focusing on short-term forecasting (RMSE) and capturing long-term behavior (invariant statistics). In addition, we would like to investigate and compare training methods [22–24] across different architectures.

**Broader impacts.** While better emulators for chaotic dynamics may be used in a wide range of applications, we foresee no direct negative societal impacts.

## Acknowledgments and Disclosure of Funding

The authors gratefully acknowledge the support of DOE grant DE-SC0022232, AFOSR grant FA9550-18-1-0166, and NSF grants DMS-2023109 and DMS-1925101. In addition, Peter Y. Lu gratefully acknowledges the support of the Eric and Wendy Schmidt AI in Science Postdoctoral Fellowship, a Schmidt Futures program.

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

# A Additional Discussion

## A.1 Contrastive feature learning vs. physics-informed optimal transport

Our two proposed approaches both aim to train neural operators to preserve the invariant measures of chaotic attractors. Ultimately, both methods have strong performance according to a variety of metrics, but the contrastive learning approach requires no prior physical knowledge and is often faster. Both approaches have significant advantages over the standard approach of minimizing RMSE, which fails to preserve important statistical characteristics of the system.

The primary conceptual difference between our approaches is that the optimal transport approach relies on prior domain knowledge, while the contrastive learning approach learns from the data alone. Specifically, the physics-informed optimal transport approach requires a choice of summary statistics, so it is much more dependent on our prior knowledge about the system and its chaotic attractor. Contrastive feature learning does not require a choice of statistics and instead learns useful invariant statistics on its own—although it may still be useful to have known relevant statistics to use as evaluation metrics for hyperparameter tuning.

The neural operators trained using the optimal transport loss perform the best on the $L_1$ histogram distance of the summary statistics $\mathbf{S}$ (Table 1 and 2), which is unsurprising given that the optimal transport loss specifically matches the distribution of $\mathbf{S}$. If we look at a different evaluation metric, e.g. the energy spectrum error or leading LE, the contrastive learning loss, in some cases, performs better even without prior knowledge. This suggests that contrastive feature learning may be capturing a wider range or different set of invariant statistics. We also see that the quality of the emulator degrades if we choose less informative statistics (Appendix B.3).

## A.2 Interactions and trade-offs between short-term prediction and long-term statistics

Comparing our results in Tables 1 and 2, we find that emulators trained using our approaches perform significantly better on the long-term evaluation metrics (e.g. $L_1$ histogram distance and energy spectrum error) by trading off a bit of performance in terms of short-term RMSE (Tables 11 and 12). This suggests that training an emulator to model chaotic dynamics using purely RMSE can result in overfitting and generally a poor model of long-term dynamics. However, we still believe that RMSE is a useful part of the loss, even for long-term evaluation metrics, since it is the only term that is directly enforcing the short-term dynamics. In some cases, we find that the invariant statistics $\hat{S}_{\mathcal{A}}$ of trajectories generated by our trained neural operators are closer to the true statistics $S_{\mathcal{A}}$ than the statistics directly computed using the noisy training data $\tilde{S}_{\mathcal{A}}$, i.e., $|\hat{S}_{\mathcal{A}} - S_{\mathcal{A}}| < |\tilde{S}_{\mathcal{A}} - S_{\mathcal{A}}|$, despite being trained on the noisy data (Table 3). We attribute this result to the RMSE component of our losses, which provides additional regularization by enforcing the short-term dynamics, combined with the inductive biases of the neural operator architectures, which are designed for modeling PDEs.

| $r$ | Training | Histogram Error $|\hat{S}_{\mathcal{A}} - S_{\mathcal{A}}| \downarrow$ | Histogram Error $|\tilde{S}_{\mathcal{A}} - S_{\mathcal{A}}| \downarrow$ |
|---|---|---|---|
| 0.1 | $\ell_{\text{RMSE}}$ | 0.056 (0.051, 0.062) | 0.029 (0.023, 0.036) |
| | $\ell_{\text{OT}} + \ell_{\text{RMSE}}$ | **0.029** (0.027, 0.032) | 0.029 (0.023, 0.036) |
| | $\ell_{\text{CL}} + \ell_{\text{RMSE}}$ | 0.033 (0.029, 0.037) | 0.029 (0.023, 0.036) |
| 0.2 | $\ell_{\text{RMSE}}$ | 0.130 (0.118, 0.142) | 0.101 (0.086, 0.128) |
| | $\ell_{\text{OT}} + \ell_{\text{RMSE}}$ | **0.039** (0.035, 0.042) | 0.101 (0.086, 0.128) |
| | $\ell_{\text{CL}} + \ell_{\text{RMSE}}$ | 0.073 (0.066, 0.080) | 0.101 (0.086, 0.128) |
| 0.3 | $\ell_{\text{RMSE}}$ | 0.215 (0.204, 0.234) | 0.213 (0.190, 0.255) |
| | $\ell_{\text{OT}} + \ell_{\text{RMSE}}$ | **0.057** (0.052, 0.064) | 0.213 (0.190, 0.255) |
| | $\ell_{\text{CL}} + \ell_{\text{RMSE}}$ | 0.132 (0.111, 0.151) | 0.213 (0.190, 0.255) |

Table 3: **Histogram error of neural operator predictions vs. noisy training data on Lorenz-96.** Averaging over 200 testing instances with varying $\phi^{(n)}$, we compared the $L_1$ histogram error of the predicted dynamics from the neural operator $|\hat{S}_{\mathcal{A}} - S_{\mathcal{A}}|$ with the histogram error of the raw noisy training data $|\tilde{S}_{\mathcal{A}} - S_{\mathcal{A}}|$. This shows that, even though the neural operator is trained on the noisy data, the statistics of the predicted dynamics are often better than those computed directly from the noisy data. Here, $S_{\mathcal{A}}$ is computed from the noiseless ground truth data.

# B Additional Experiments and Evaluation Metrics

We have performed several additional experiments that act as points of comparison, help us better understand the behavior of our methods under a variety of conditions, and provide useful insights for future applications of our approaches.

## B.1 Denoising with Gaussian blurring

Gaussian blurring, often used as a denoising technique for images, employs a Gaussian distribution to establish a convolution matrix that's applied to the original image. The fundamental idea involves substituting the noisy pixel with a weighted average of surrounding pixel values. A key hyperparameter in Gaussian blurring is the standard deviation of the Gaussian distribution. When the standard deviation approaches zero, it fundamentally indicates the absence of any blur. Under such circumstances, the Gaussian function collapses to a single point, leading to the elimination of the blur effect. In Table 4, we present the results from applying Gaussian blurring to noisy data during training based solely on RMSE. Despite the effectiveness of the widely adopted denoising approach, our findings indicate that Gaussian blurring may not be ideally suited for our purpose of emulating dynamics. This is primarily because significant invariant statistics might be strongly correlated with certain high-frequency signals that could be affected by the blurring preprocessing.

| Training | Histogram Error ↓ | Energy Spec. Error↓ | Leading LE Error ↓ |
|---|---|---|---|
| $\ell_{\mathrm{RMSE}}$ ($\sigma_b = 0.1$) | **0.390** (0.326, 0.556) | **0.290** (0.226, 0.402) | **0.098** (0.069, 0.127) |
| $\ell_{\mathrm{RMSE}}$ ($\sigma_b = 0.5$) | 1.011 (0.788, 1.264) | 0.493 (0.379, 0.623) | **0.098** (0.041, 0.427) |
| $\ell_{\mathrm{RMSE}}$ | **0.390** (0.325, 0.556) | **0.290** (0.225, 0.402) | 0.101 (0.069, 0.122) |

Table 4: **Emulator performance with Gaussian blurring on Kuramoto–Sivashinsky data with noise scale** $r = 0.3$. Averaging over 200 testing instances with varying $\phi^{(n)}$, we show the performance of (1) the application of Gaussian blurring as a preliminary denoising effort with a small standard deviation ($\sigma_b = 0.1$); (2) the application of Gaussian blurring with a larger standard deviation ($\sigma_b = 0.5$); and (3) training purely on RMSE without any blurring preprocessing. The results suggest that the application of Gaussian blurring might further degrade the results, as the high-frequency signals associated with invariant statistics can be lost.

## B.2 Sobolev norm baseline

We recognize that there are alternative methods that strive to capture high-frequency signals by modifying training objectives. For instance, the Sobolev norm, which combines data and its derivatives, has been found to be quite effective in capturing high-frequency signals [16, 49]. However, its effectiveness can be significantly curtailed in a noisy environment, especially when noise is introduced to a high-frequency domain, as minimizing the Sobolev norm then fails to accurately capture relevant statistics, as shown in Tables 5 and 6.

| Training | Histogram Error ↓ | Energy Spec. Error ↓ | Leading LE Error ↓ | FD Error ↓ |
|---|---|---|---|---|
| $\ell_{\mathrm{RMSE}}$ | 0.215 (0.204, 0.234) | 0.291 (0.280, 0.305) | 0.440 (0.425, 0.463) | 3.580 (2.333, 4.866) |
| $\ell_{\mathrm{Sobolev}} + \ell_{\mathrm{dissaptive}}$ | 0.246 (0.235, 0.255) | 0.325 (0.341, 0.307) | 0.487 (0.456, 0.545) | 4.602 (3.329, 6.327) |
| $\ell_{\mathrm{OT}} + \ell_{\mathrm{RMSE}}$ | **0.057** (0.052, 0.064) | **0.123** (0.116, 0.135) | 0.084 (0.062, 0.134) | 3.453 (2.457, 4.782) |
| $\ell_{\mathrm{CL}} + \ell_{\mathrm{RMSE}}$ | 0.132 (0.111, 0.151) | 0.241 (0.208, 0.285) | **0.064** (0.045, 0.091) | **1.894** (0.942, 3.108) |

Table 5: **Emulator performance (including Sobolev norm loss) on Lorenz-96 data with noise scale** $r = 0.3$. The median (25th, 75th percentile) of the evaluation metrics are computed on 200 Lorenz-96 test instances (each with 1500 time steps) for the neural operator trained with (1) only RMSE loss $\ell_{\mathrm{RMSE}}$; (2) Sobolev norm loss with dissipative regularization $\ell_{\mathrm{Sobolev}} + \ell_{\mathrm{dissaptive}}$; (3) optimal transport (OT) and RMSE loss $\ell_{\mathrm{OT}} + \ell_{\mathrm{RMSE}}$; and (4) contrastive learning (CL) and RMSE loss $\ell_{\mathrm{CL}} + \ell_{\mathrm{RMSE}}$.

| Training | Histogram Error ↓ | Energy Spec. Error ↓ | Leading LE Error ↓ |
|---|---|---|---|
| $\ell_{\text{RMSE}}$ | 0.390 (0.325, 0.556) | 0.290 (0.225, 0.402) | 0.101 (0.069, 0.122) |
| $\ell_{\text{Sobolev}} + \ell_{\text{dissipative}}$ | 0.427 (0.289, 0.616) | 0.237 (0.204, 0.315) | **0.023** (0.012, 0.047) |
| $\ell_{\text{OT}} + \ell_{\text{RMSE}}$ | **0.172** (0.146, 0.197) | 0.211 (0.188, 0.250) | 0.094 (0.041, 0.127) |
| $\ell_{\text{CL}} + \ell_{\text{RMSE}}$ | 0.193 (0.148, 0.247) | **0.176** (0.130, 0.245) | 0.108 (0.068, 0.132) |

Table 6: **Emulator performance (including Sobolev norm loss) on Kuramoto–Sivashinsky data with noise scale $r = 0.3$.** The median (25th, 75th percentile) of the evaluation metrics are computed on 200 Kuramoto–Sivashinsky test instances (each with 1000 time steps) for the neural operator trained with (1) only RMSE loss $\ell_{\text{RMSE}}$; (2) Sobolev norm loss with dissipative regularization $\ell_{\text{Sobolev}} + \ell_{\text{dissaptive}}$; (3) optimal transport (OT) and RMSE loss $\ell_{\text{OT}} + \ell_{\text{RMSE}}$; and (4) contrastive learning (CL) and RMSE loss $\ell_{\text{CL}} + \ell_{\text{RMSE}}$.

### B.3 Optimal transport: reduced set of summary statistics

For our optimal transport approach, we test a reduced set of summary statistics, which shows how the quality of the summary statistic affects the performance of the method (Table 7). With an informative summary statistic, we find even a reduced set can still be helpful but, for a non-informative statistic, the optimal transport method fails as expected.

| Training statistics | Histogram Error ↓ | Energy Spec. Error ↓ | Leading LE Error ↓ | FD Error ↓ |
|---|---|---|---|---|
| **S** (full) | **0.057** (0.052, 0.064) | **0.123** (0.116, 0.135) | 0.084 (0.062, 0.134) | 3.453 (2.457, 4.782) |
| $\mathbf{S}_1$ (partial) | 0.090 (0.084, 0.098) | 0.198 (0.189, 0.208) | 0.263 (0.217, 0.323) | 3.992 (2.543, 5.440) |
| $\mathbf{S}_2$ (minimum) | 0.221 (0.210, 0.234) | 0.221 (0.210, 0.230) | 0.276 (0.258, 0.291) | **3.204** (2.037, 4.679) |

Table 7: **Emulator performance for different choices of summary statistics on Lorenz-96 data with noise scale $r = 0.3$.** Each neural operator was trained using the optimal transport and RMSE loss using (1) full statistics $\mathbf{S}(\mathbf{u}) := \{\frac{du_i}{dt}, (u_{i+1} - u_{i-2})u_{i-1}, u_i\}$; (2) partial statistics $\mathbf{S}_1(\mathbf{u}) := \{(u_{i+1} - u_{i-2})u_{i-1}\}$; or (3) minimum statistics $\mathbf{S}_2(\mathbf{u}) := \{\bar{\mathbf{u}}\}$, where $\bar{\mathbf{u}}$ is the spatial average.

### B.4 Contrastive learning: reduced environment diversity

For our contrastive learning approach, we test a multi-environment setting with reduced data diversity and find that the contrastive method still performs well under the reduced conditions (Table 8), which demonstrates robustness.

| Training | Histogram Error ↓ | Energy Spec. Error ↓ | Leading LE Error ↓ | FD Error ↓ |
|---|---|---|---|---|
| $\ell_{\text{RMSE}}$ | 0.255 (0.248, 0.263) | 0.307 (0.302, 0.315) | 0.459 (0.743, 2.746) | 3.879 (2.456, 5.076) |
| $\ell_{\text{OT}} + \ell_{\text{RMSE}}$ | **0.055** (0.050, 0.061) | **0.124** (0.116, 0.131) | 0.080 (0.045, 0.109) | 4.015 (2.401, 5.225) |
| $\ell_{\text{CL}} + \ell_{\text{RMSE}}$ | 0.130 (0.111, 0.152) | 0.193 (0.183, 0.200) | **0.031** (0.014, 0.053) | **1.747** (0.792, 2.939) |

Table 8: **Emulator performance with reduced environment diversity (i.e. narrower parameter range) on Lorenz-96 data with noise level $r = 0.3$.** Averaging over 200 testing instances, we show the performance of training the neural operator with (1) only RMSE loss $\ell_{\text{RMSE}}$; (2) optimal transport (OT) and RMSE loss $\ell_{\text{OT}} + \ell_{\text{RMSE}}$; and (3) contrastive learning (CL) and RMSE loss $\ell_{\text{CL}} + \ell_{\text{RMSE}}$. We shrink the parameter range for generating the dataset from $[10, 18]$ to $[16, 18]$.

## B.5 Maximum mean discrepancy (MMD) vs. optimal transport loss

We also implement a variant of our optimal transport approach that uses maximum mean discrepancy (MMD) as a distributional distance rather than the Sinkhorn divergence. Using the same set of summary statistics, we find that MMD does not perform as well as our optimal transport loss for training emulators (Table 9).

| Training | Histogram Error ↓ | Energy Spec. Error ↓ | Leading LE Error ↓ |
|---|---|---|---|
| $\ell_{\mathrm{RMSE}}$ | 0.390 (0.325, 0.556) | 0.290 (0.225, 0.402) | 0.101 (0.069, 0.122) |
| $\ell_{\mathrm{MMD}} + \ell_{\mathrm{RMSE}}$ | 0.245 (0.218, 0.334) | 0.216 (0.186, 0.272) | 0.101 (0.058, 0.125) |
| $\ell_{\mathrm{OT}} + \ell_{\mathrm{RMSE}}$ | **0.172** (0.146, 0.197) | 0.211 (0.188, 0.250) | **0.094** (0.041, 0.127) |
| $\ell_{\mathrm{CL}} + \ell_{\mathrm{RMSE}}$ | 0.193 (0.148, 0.247) | **0.176** (0.130, 0.245) | 0.108 (0.068, 0.132) |

Table 9: **Emulator performance (including MMD loss) on Kuramoto–Sivashinsky data with noise scale** $r = 0.3$**.** The median (25th, 75th percentile) of the evaluation metrics are computed on 200 Kuramoto–Sivashinsky test instances (each with 1000 time steps) for the neural operator trained with (1) only RMSE loss $\ell_{\mathrm{RMSE}}$; (2) maximum mean discrepency (MMD) and RMSE loss $\ell_{\mathrm{MMD}} + \ell_{\mathrm{RMSE}}$; (3) optimal transport (OT) and RMSE loss $\ell_{\mathrm{OT}} + \ell_{\mathrm{RMSE}}$; and (4) contrastive learning (CL) and RMSE loss $\ell_{\mathrm{CL}} + \ell_{\mathrm{RMSE}}$.

## B.6 Additional Lyapunov spectrum evaluation metrics

In the table 10, we evaluated the results of Lorenz 96 on Lyapunov spectrum error rates and the total number of positive Lyapunov exponents error rates. For the Lyapunov spectrum error, we report the sum of relative absolute errors across the full spectrum: $\sum_i^d |\hat{\lambda}_i - \lambda_i| / \lambda_i$, where $\lambda_i$ is the $i$-th Lyapunov exponent and $d$ is the dimension of the dynamical state. As suggested by [50], we also compare the number of positive Lyapunov exponents (LEs) as an additional statistic to measure the complexity of the chaotic dynamics. We compute the absolute error in the number of positive LEs $\sum_i^d |\mathbf{1}(\hat{\lambda}_i > 0 - \mathbf{1}(\lambda_i > 0)|$.

| $r$ | Training | Leading LE Error ↓ | Lyapunov Spectrum Error ↓ | Total number of positive LEs Error ↓ |
|---|---|---|---|---|
| | $\ell_{\mathrm{RMSE}}$ | **0.013** (0.006, 0.021) | 0.265 (0.110, 0.309) | 0.500 (0.000, 1.000) |
| 0.1 | $\ell_{\mathrm{OT}} + \ell_{\mathrm{RMSE}}$ | 0.050 (0.040, 0.059) | 0.248 (0.168, 0.285) | **0.000** (0.000, 1.000) |
| | $\ell_{\mathrm{CL}} + \ell_{\mathrm{RMSE}}$ | 0.065 (0.058, 0.073) | **0.227** (0.164, 0.289) | **0.000** (0.000, 1.000) |
| | $\ell_{\mathrm{RMSE}}$ | 0.170 (0.156, 0.191) | 0.612 (0.522, 0.727) | 4.000 (4.000, 5.000) |
| 0.2 | $\ell_{\mathrm{OT}} + \ell_{\mathrm{RMSE}}$ | 0.016 (0.006, 0.030) | 0.513 (0.122, 0.590) | **3.000** (2.000, 3.000) |
| | $\ell_{\mathrm{CL}} + \ell_{\mathrm{RMSE}}$ | **0.012** (0.006, 0.018) | **0.459** (0.138, 0.568) | **3.000** (2.000, 3.000) |
| | $\ell_{\mathrm{RMSE}}$ | 0.440 (0.425, 0.463) | 0.760 (0.702, 0.939) | 7.000 (7.000, 8.000) |
| 0.3 | $\ell_{\mathrm{OT}} + \ell_{\mathrm{RMSE}}$ | 0.084 (0.062, 0.134) | **0.661** (0.572, 0.746) | **5.000** (4.000, 6.000) |
| | $\ell_{\mathrm{CL}} + \ell_{\mathrm{RMSE}}$ | **0.064** (0.045, 0.091) | 0.654 (0.558, 0.780) | 6.000 (5.000, 6.000) |

Table 10: **Emulator performance on Lyapunov spectrum metrics for Lorenz-96 data.** The median (25th, 75th percentile) of the Lyapunov spectrum metrics are computed on 200 Lorenz-96 test instances (each with 1500 time steps) for the neural operator trained with (1) only RMSE loss $\ell_{\mathrm{RMSE}}$; (2) optimal transport (OT) and RMSE loss $\ell_{\mathrm{OT}} + \ell_{\mathrm{RMSE}}$; and (3) contrastive learning (CL) and RMSE loss $\ell_{\mathrm{CL}} + \ell_{\mathrm{RMSE}}$. In the presence of high noise, OT and CL give lower relative errors on the leading Lyapunov exponent (LE). When evaluating the full Lyapunov spectrum, OT and CL show significant advantages than the baseline. In addition, the lower absolute errors of the total number of the positive Lyapunov exponents (LEs) suggest that OT and CL are able to match the complexity of the true chaotic dynamics.

### B.7  1-step RMSE evaluation results

Evaluating on 1-step RMSE only shows short-term prediction performance and is not an informative evaluation metric for long-term behavior. Here, we report the 1-step RMSE (Tables 11 and 12) to show that training using our approaches retains similar 1-step RMSE results while significantly improving on long-term statistical metrics (Tables 1 and 2). See Appendix A.2 for additional discussion.

| $r$ | Training | RMSE ↓ |
|---|---|---|
| | $\ell_{\mathrm{RMSE}}$ | 0.107 (0.105, 0.109) |
| 0.1 | $\ell_{\mathrm{OT}} + \ell_{\mathrm{RMSE}}$ | 0.108 (0.105, 0.110) |
| | $\ell_{\mathrm{CL}} + \ell_{\mathrm{RMSE}}$ | 0.109 (0.108, 0.113) |
| | $\ell_{\mathrm{RMSE}}$ | 0.202 (0.197, 0.207) |
| 0.2 | $\ell_{\mathrm{OT}} + \ell_{\mathrm{RMSE}}$ | 0.207 (0.202, 0.212) |
| | $\ell_{\mathrm{CL}} + \ell_{\mathrm{RMSE}}$ | 0.214 (0.203, 0.218) |
| | $\ell_{\mathrm{RMSE}}$ | 0.288 (0.282, 0.296) |
| 0.3 | $\ell_{\mathrm{OT}} + \ell_{\mathrm{RMSE}}$ | 0.301 (0.293, 0.307) |
| | $\ell_{\mathrm{CL}} + \ell_{\mathrm{RMSE}}$ | 0.312 (0.302,0.316) |

Table 11: **1-step RMSE performance on Lorenz-96 data.** The median (25th, 75th percentile) of the Lyapunov spectrum metrics are computed on 200 Lorenz-96 test instances (each with 1500 time steps) for the neural operator trained with (1) only RMSE loss $\ell_{\mathrm{RMSE}}$; (2) optimal transport (OT) and RMSE loss $\ell_{\mathrm{OT}} + \ell_{\mathrm{RMSE}}$; and (3) contrastive learning (CL) and RMSE loss $\ell_{\mathrm{CL}} + \ell_{\mathrm{RMSE}}$. We see comparable performance on short-term 1-step RMSE.

| Training | RMSE ↓ |
|---|---|
| $\ell_{\mathrm{RMSE}}$ | 0.373 (0.336, 0.421) |
| $\ell_{\mathrm{OT}} + \ell_{\mathrm{RMSE}}$ | 0.381 (0.344, 0.430) |
| $\ell_{\mathrm{CL}} + \ell_{\mathrm{RMSE}}$ | 0.402 (0.364, 0.452) |

Table 12: **1-step RMSE performance on Kuramoto–Sivashinsky data with noise scale $r = 0.3$.** The median (25th, 75th percentile) of the evaluation metrics are computed on 200 Kuramoto–Sivashinsky test instances (each with 1000 time steps) for the neural operator trained with (1) only RMSE loss $\ell_{\mathrm{RMSE}}$; (2) optimal transport (OT) and RMSE loss $\ell_{\mathrm{OT}} + \ell_{\mathrm{RMSE}}$; and (3) contrastive learning (CL) and RMSE loss $\ell_{\mathrm{CL}} + \ell_{\mathrm{RMSE}}$. We again see comparable performance on short-term 1-step RMSE.

## C  Implementation Details

### C.1  Evaluation metrics

To evaluate the long-term statistical behavior of our trained neural operators, we run the neural operator in a recurrent fashion for 1500 (Lorenz-96) or 1000 (Kuramoto–Sivashinsky) time steps and then compute our long-term evaluation metrics on this autonomously generated time-series.

**Histogram error.** For distributional distance with a pre-specified statistics $\mathbf{S}$, we first compute the histogram of the invariant statistics $\mathbf{H}(\mathbf{S}) = \{(\mathbf{S}_1, c_1), (\mathbf{S}_2, c_2), \dots, (\mathbf{S}_B, c_B)\}$, where $\mathbf{H}$ represents the histogram, and the values of the bins are denoted by $\mathbf{S}_b$ with their corresponding frequencies $c_b$. We then define the $L_1$ histogram error as:

$$\mathrm{Err}(\hat{\mathbf{H}}, \mathbf{H}) := \sum_{b=1}^{B} \|c_b - \hat{c}_b\|_1. \tag{21}$$

Note that for a fair comparison across all our experiments, we use the rule of thumb—the square root rule to decide the number of bins.

**Energy spectrum error.** We compute the relative mean absolute error of the energy spectrum—the squared norm of the spatial FFT $\mathcal{F}[\mathbf{u}_t]$—averaged over time:

$$\frac{1}{T} \sum_{\mathbf{u}_t, \hat{\mathbf{u}}_t \in \mathbf{U}_{1:T}, \hat{\mathbf{U}}_{1:T}} \frac{\||\mathcal{F}[\mathbf{u}_t]|^2 - |\mathcal{F}[\hat{\mathbf{u}}_t]|^2\|_1}{\||\mathcal{F}[\mathbf{u}_t]|^2\|_1}. \tag{22}$$

**Leading LE error.** The leading Lyapunov exponent (LE) is a dynamical invariant that measures how quickly the chaotic system becomes unpredictable. For the leading LE error, we report the relative absolute error $|\hat{\lambda} - \lambda|/|\lambda|$ between the model and the ground truth averaged over the test set. We adapted the Julia DynamicalSystem.jl package to calculate the leading LE.

**FD error.** The fractal dimension (FD) is a characterization of the dimension of the attractor. We report the absolute error $|\hat{D} - D|$ between the estimated FD of the model and the ground truth averaged over the test set. We use the Julia DynamicalSystem.jl package for calculating the fractal dimension.

**RMSE.** We use 1-step relative RMSE $\|\mathbf{u}_{t+\Delta t} - \hat{g}_\theta(\mathbf{u}_t, \phi)\|_2/\|\mathbf{u}_{t+\Delta t}\|_2$ to measure short-term prediction accuracy.

## C.2  Time complexity

The optimal transport approach relies on the Sinkhorn algorithm which scales as $\mathcal{O}(n^2 \log n)$ for comparing two distributions of $n$ points each (Theorem 2 in [51]). In our experiments, we use $n = 6000$ to $n = 25600$ points with no issues, so this approach scales relatively well. The contrastive learning approach requires pretraining but is even faster during emulator training since it uses a fixed, pre-trained embedding network.

|  | Training | Encoder | Operator | Total |
|---|---|---|---|---|
| | $\ell_{\text{RMSE}}$ | – | 20 | 20 |
| Lorenz-96 | $\ell_{\text{OT}} + \ell_{\text{RMSE}}$ | – | 51 | 51 |
| | $\ell_{\text{CL}} + \ell_{\text{RMSE}}$ | 22 | 27 | 49 |
| | $\ell_{\text{RMSE}}$ | – | 55 | 55 |
| Kuramoto–Sivashinsky | $\ell_{\text{OT}} + \ell_{\text{RMSE}}$ | – | 262 | 262 |
| | $\ell_{\text{CL}} + \ell_{\text{RMSE}}$ | 26 | 56 | 82 |

Table 13: **Empirical training time.** Training time (minutes) with 4 GPUs for 60-dimensional Lorenz-96 and 256-dimensional Kuramoto–Sivashinsky.

## C.3  Training of the encoder

**Evaluation rule.** A standard procedure for assessing the performance of an encoder trained with Noise Contrastive Estimation (InfoNCE) loss in an unsupervised manner [32–34], is employing the top-1 accuracy metric. This measures how effectively similar items are positioned closer to each other compared to dissimilar ones in the embedded space. While this evaluation measure is commonly employed in downstream tasks that focus on classification, it is suitably in line with our goals. We aim to learn an encoder that can differentiate whether sequences of trajectories are sampled from different attractors, each characterized by distinct invariant statistics. Therefore, the use of Top-1 accuracy to evaluate if two sequences originate from the same trajectory serves our purpose effectively. And we utilize this metric during our training evaluations to assess the performance of our encoder.

The formal definition of Top-1 accuracy requires us to initially define the softmax function $\sigma$, which is used to estimate the likelihood that two samples from different time windows originate from the same trajectory,

$$\sigma\left(f_\psi(\mathbf{U}_{J:J+K}^{(j)}); f_\psi(\mathbf{U}_{I:I+K}^{(n)})\right) = \frac{\exp\left(\langle f_\psi(\mathbf{U}_{I:I+K}^{(n)}), f_\psi(\mathbf{U}_{J:J+K}^{(j)})\rangle\right)}{\sum_{m=1}^{N}\left[\exp\left(\langle f_\psi(\mathbf{U}_{I:I+K}^{(n)}), f_\psi(\mathbf{U}_{H:H+K}^{(m)})\rangle\right)\right]}. \tag{23}$$

Using the softmax function to transform the encoder's output into a probability distribution, we then define Top-1 accuracy as:

$$\text{Acc}_1 = \frac{1}{N}\sum_{n=1}^{N}\mathbf{I}\left\{n = \arg\max_j \sigma\left(f_\psi(\mathbf{U}_{J:J+K}^{(j)}); f_\psi(\mathbf{U}_{I:I+K}^{(n)})\right)\right\}, \tag{24}$$

where $\mathbf{1}(\cdot)$ is the indicator function representing whether two most close samples in the feature space comes from the same trajectory or not.

**Training hyperparameters.** We use the ResNet-34 as the backbone of the encoder, throughout all experiments, we train the encoder using the AdamW optimization algorithm [52], with a weight decay of $10^{-5}$, and set the training duration to 2000 epochs.

For the temperature value $\tau$ balancing the weights of difficult and easy-to-distinguish samples in contrastive learning, we use the same warm-up strategy as in [10]. Initially, we start with a relatively low $\tau$ value (0.3 in our experiments) for the first 1000 epochs. This ensures that samples that are difficult to distinguish get large gradients. Subsequently, we incrementally increase $\tau$ up to a specified value (0.7 in our case), promoting the grouping of similar samples within the embedded space. From our empirical observations, we have found that this approach leads to an improvement in Top-1 accuracy in our experiments.

The length of the sequence directly influences the Top-1 accuracy. Considering that the sample length $L$ of the training data $\{\mathbf{U}_{0:L}^{(n)}\}_{n=1}^N$ is finite, excessively increasing the crop length $K$ can have both advantages and disadvantages. On one hand, it enables the encoder to encapsulate more information; on the other hand, it could lead to the failure of the encoder's training. This is likely to occur if two samples, i.e., $f_\psi(\mathbf{U}_{I:I+K}^{(n)})$ and $f_\psi(\mathbf{U}_{J:J+K}^{(n)})$, from the same trajectory overlap excessively, inhibiting the encoder from learning a meaningful feature space. With this consideration, we've empirically chosen the length $K$ of subsequences for the encoder to handle to be approximately 5% of the total length $T$.

### C.4 Lorenz-96

**Data generation.** To better align with realistic scenarios, we generate our training data with random initial conditions drawn from a normal distribution. For the purpose of multi-environment learning, we generate 2000 trajectories for training. Each of these trajectories has a value of $\phi^{(n)}$ randomly sampled from a uniform distribution ranging from 10.0 to 18.0. We discretize these training trajectories into time steps of $dt = 0.1$ over a total time of $t = 205s$, yielding 2050 discretized time steps. Moreover, in line with the setup used in [2, 11], we discard the initial 50 steps, which represent the states during an initial transient period.

**Training hyperparameters.** We determine the roll-out step during training (i.e., $h$ and $h_{\mathrm{RMSE}}$) via the grid search from the set of values $\{1, 2, 3, 4, 5\}$. Though it is shown in some cases that a larger roll-out number help improve the results [18], we have not observed that in our training. We hypothesized that this discrepancy may be due to our dataset being more chaotic compared to typical cases. Consequently, to ensure a fair comparison and optimal outcomes across all experiments, we have decided to set $h$ and $h_{\mathrm{RMSE}}$ to 1.

In the case of the optimal transport loss[2], described in Section 3.1, the blur parameter $\gamma$ governs the equilibrium between faster convergence speed and the precision of the Wasserstein distance approximation. We determined the value of $\gamma$ through a grid search, examining values ranging from $\{0.01, 0.02, \ldots, 0.20\}$. Similarly, we decided the weights of the optimal transport loss $\alpha$ through a grid search, exploring values from $\{0.5, 1, 1.5, \ldots, 3.0\}$. For the experiments conducted using Lorenz-96, we selected $\alpha = 3$ and $\gamma = 0.02$.

In the context of the feature loss, the primary hyperparameter we need to consider is its weights, represented as $\lambda$. Given that we do not have knowledge of the invariant statistics $\mathbf{S}$ during the validation phase, we adjust $\lambda$ according to a specific principle: our aim is to reduce the feature loss $\ell_{\mathrm{feature}}$ (as defined in Eqn. 17) as long as it does not adversely affect the RMSE beyond a predetermined level (for instance, 10%) when compared with the baseline model, which is exclusively trained on RMSE. As illustrated in Figure 4, we adjust the values of $\lambda$ in a systematic grid search from $\{0.2, 0.4, 0.6, 0.8, 1.0, 1.2\}$. From our observations during the validation phase, we noted that the feature loss was at its lowest with an acceptable RMSE (which is lower than 110% compared to the baseline) when $\lambda = 0.8$. Therefore, we have reported our results with $\lambda$ set at 0.8.

**Results visualization.** We present more results with varying noise levels in Figures 5, 6, 7.

---

[2] https://www.kernel-operations.io/geomloss/

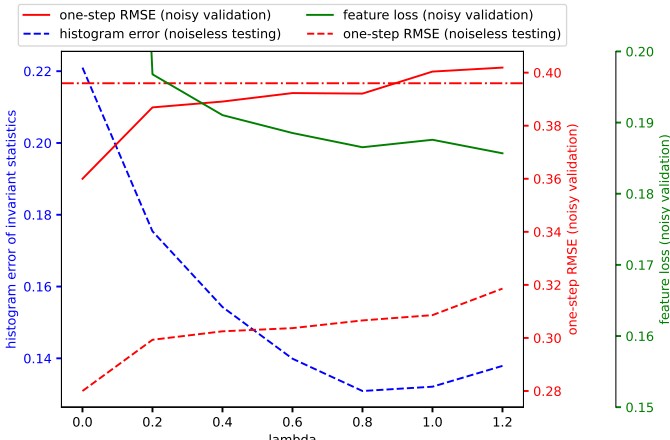

Figure 4: **The trend of feature loss with its weight $\lambda$ when the scale of noise is** $r = 0.3$**.** The solid lines in the figure represent the evaluation metrics during the validation phase, comparing the outputs of the neural operator to the noisy data. In contrast, the dashed lines represent the actual metrics we are interested in, comparing the outputs of the neural operator to the clean data and calculating the error of the invariant statistics. In addition, the horizontal solid dashed line correspond to the bar we set for the RMSE, i.e., $110\%$ of the RMSE when $\lambda = 0$. We observe that, (1) with the increase of $\lambda$ from 0, the feature loss decreases until $\lambda$ reaches 1.0. (2) The RMSE generally increases with the increase of $\lambda$. (3) The unseen statistical error generally decreases with the increase of $\lambda$. We reported the results when $\lambda = 0.8$ as our final result, since the further increase of $\lambda$ does not bring further benefit in decreasing the feature loss, and the result remains in an acceptable range in terms of RMSE.

### C.5    Kuramoto–Sivashinsky

**Data generation.**  In order to ascertain that we are operating within a chaotic regime, we set the domain size $L = 50$ and the spatial discretization grids number $d = 256$. We select initial conditions randomly from a uniform range between $[-\pi, \pi]$. A fourth-order Runge-Kutta method was utilized to perform all simulations. We generate 2000 trajectories for training, with each $\phi^{(n)}$ being randomly chosen from a uniform distribution within the interval $[1.0, 2.6]$.

**Training hyperparameters.**  In the case of the optimal transport loss, similar to the discussion in C.4, we search the blur value $\gamma$ and the weights of loss $\alpha$ from a grid search. And in our experiment, we set $\alpha = 3$ and $\gamma = 0.05$. In the case of the feature loss, we again adopt the same rule for deciding the value of $\lambda$, where we compare the trend of feature loss with the relative change of RMSE of the baseline and choose to report the results with the lowest feature loss and acceptable RMSE increase ($110\%$ compared to the baseline when $\lambda = 0$).

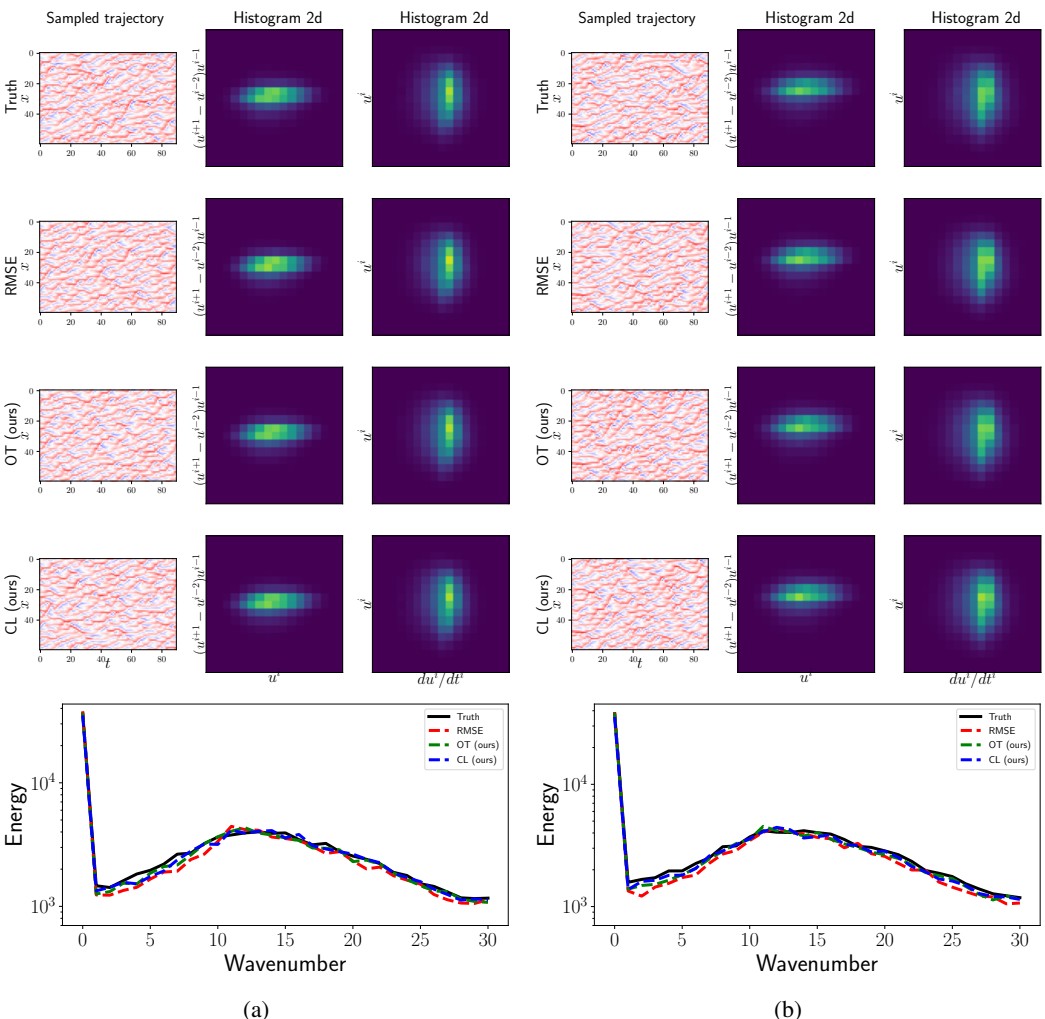

Figure 5: **Visualization of the predictions when the noise Level** $r = 0.1$. We evaluate our method by comparing them to the baseline that is trained solely using RMSE. For two different instances (a) and (b), we visualize the visual comparison of the predicted dynamics (left), two-dimensional histograms of relevant statistics (middle and right). We notice that, with the minimal noise, the predictions obtained from all methods look statistically consistent to the true dynamics.

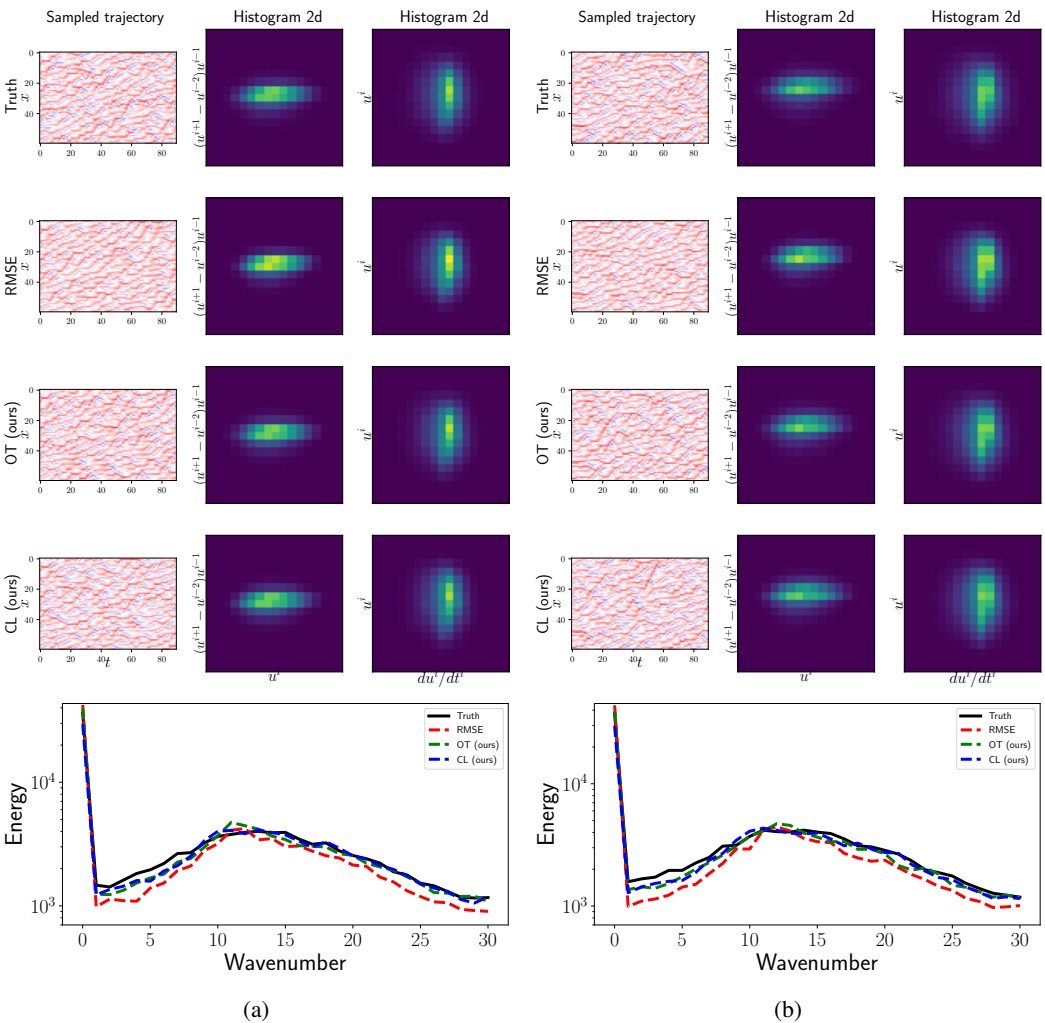

Figure 6: **Visualization of the predictions when the noise level** $r = 0.2$. We evaluate our method by comparing them to the baseline that is trained solely using RMSE. For two different instances (a) and (b), we visualize the visual comparison of the predicted dynamics (left), two-dimensional histograms of relevant statistics (middle and right). We observe that as the noise level escalates, the degradation of performance in the results of RMSE is more rapid compared to our method, which employs optimal transport and feature loss during training.

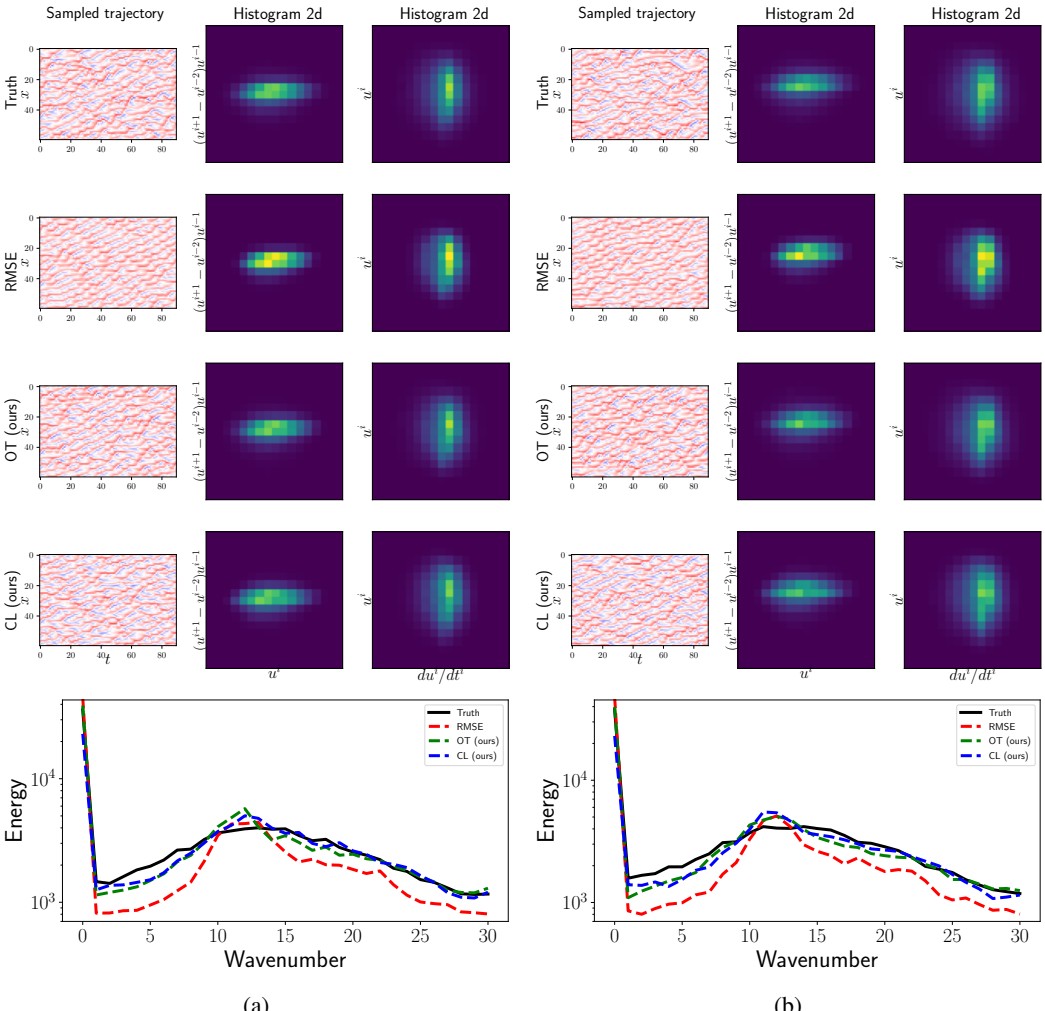

Figure 7: **Visualization of the predictions when the noise level** $r = 0.3$. We evaluate our method by comparing them to the baseline that is trained solely using RMSE. For two different instances (a) and (b), we visualize the visual comparison of the predicted dynamics (left), two-dimensional histograms of relevant statistics (middle and right). We find that, under higher levels of noise, the RMSE results exhibit fewer negative values, as indicated by the blue stripes in the predicted dynamics. This is further confirmed by the energy spectrum, which clearly shows that the RMSE results are significantly deficient in capturing high-frequency signals.

