# OpenReview forum: "Training neural operators to preserve invariant measures of chaotic attractors"
_NeurIPS.cc/2023/Conference — NeurIPS 2023 poster_

### Official Review · Reviewer_tYBd · 2023-07-03

**Soundness:** 3 good
**Presentation:** 3 good
**Contribution:** 2 fair
**Rating:** 5
**Confidence:** 3

**Summary:**

This paper mentions that MSE loss is insufficient for learning the solution operator for chaotic dynamics. Two approaches are proposed to solve the problem. One is to use sinkhorn loss, and the other is to use contrastive learning. The authors evaluate the effectiveness of the proposed methods through experiments on several physical systems.

**Strengths:**

- The problem of learning solution operators for chaotic dynamics is a challenging and interesting problem, and the authors have an interesting point of view (preserving invariant measures).
- To my knowledge, the introduction of physics-informed optimal transport and contrastive learning is a new approach.


**Weaknesses:**

- Although the problem setting is important and interesting, the proposed method seems to be a straightforward combination of existing technologies.
- There is room for improvement in the writing of the paper.
- There is no reference to previous studies on autoregressive approach.

**Questions:**

- I understood that the prediction in equation (3) is the same as the autoregressive PDE solver proposed in [R1]. It would be good to add such previous studies.

- Since the autoregressive approach performs forecasting iteratively, it seems to me that the numerical error increases in the long-term forecasting setting. Could you add a discussion on this point? Also, is it possible to forecast at arbitrary time intervals $\Delta t$ during training and testing?

- I think it would be easier to read if the description of the previous studies (optimal transport and contrastive learning) and the proposed method are separated. In section 3, I feel that the novelty of this paper is not described clearly.

- There are other ways (e.g., Maximum Mean Discrepancy) to measure the distance of a distribution besides optimal transport. It would be good to clarify the motivation for using optimal transport, and the same goes for contrastive learning.

[R1] Johannes Brandstetter, Daniel E. Worrall, Max Welling, Message Passing Neural PDE Solvers, ICLR, 2022.

**Limitations:**

It would be good to discuss in detail the advantages and disadvantages of using Optimal transport and contrastive learning.

---

> ### Author Rebuttal · Authors · 2023-08-10
>
> Thank you for your feedback!
>
> **W1**:
> > Although the problem setting is important and interesting, the proposed method seems to be a straightforward combination…
>
> Our proposed approaches are not a simple reshuffling of methods to solve a variant of the originally proposed problem (e.g. general representation learning) but rather a completely new use of contrastive learning (CL) and optimal transport (OT) to address a very different problem: training emulators of chaotic systems with good long-term behavior. Our contributions include both framing the problem and then developing new and adapting existing methods to handle this new problem domain.
>
> Our application of CL and OT is deeply motivated by our problem formulation, without which it would be impossible to justify their use in this context. Furthermore, adapting CL for this application is highly non-trivial since it is well outside of its traditional use as a general representation learning method. In the paper, we argue that CL has the right properties to learn exactly the kind of time-invariant statistics that we aim to emulate. We then show that the learned latent space can be adapted into a loss function to train emulators that successfully capture long-term behavior.
>
> The final paper will emphasize these points.
>
> **W2**:
> > There is room for improvement in the writing of the paper.
>
> Please let us know if there are any specific writing issues or sections in need of clarification.
>
> **W3/Q1**:
> > There is no reference to previous studies on autoregressive approach.
> > I understood that the prediction in equation (3) is the same as the autoregressive PDE solver… [R1].
>
> We would be happy to include additional references on emulators. Our work focuses on developing regularizers for training emulators on chaotic systems and not on autoregressive training, and we do not claim autoregressive training to be a novel contribution here.
>
> **Q2**:
> > Since the autoregressive approach performs forecasting iteratively, it seems to me that the numerical error increases in the long-term forecasting setting.
>
> The key is that, for chaotic dynamics, it does not matter what kind of emulator you choose (autoregressive or not), you will never be able to exactly predict the state of the system in the long term. That is, for any emulator, the mean squared error for long-term predictions will exponentially increase with time. That is why we focus on training emulators that capture long-term statistical behavior, which is the relevant measure of long-term forecasting performance for chaotic systems.
>
> > Also, is it possible to forecast at arbitrary time intervals…
>
> Our emulator does perform 1-step-ahead forecasting for a specific $\Delta_t$ which can be applied iteratively, but forecast accuracy at a specific time point is not the focus of our work (and indeed an impossible task for chaotic systems). A good analog of our goal would be a climate model which is unable to make an exact prediction for an arbitrary time point but does provide good statistical insights into long-term trends such as the frequency and intensity of hurricanes.
>
> **Q3**:
> > I think it would be easier to read if… the previous studies… and the proposed method are separated.
>
> In the paper, previous studies are presented in the related work (Section 1.2) and the two proposed approaches are presented in separate sections: physics-informed OT (Section 3.1) and unsupervised CL (Section 3.2).
>
> > In section 3, I feel that the novelty of this paper is not described clearly.
>
> Section 3 presents our approaches to solving the problem formulated in Section 2. The novelty of the work is discussed in our contributions statement (Section 1.1). To reiterate, we show that the standard method for training emulators is insufficient for modeling chaotic dynamics and instead propose to train the emulator to preserve the invariant measure of the chaotic attractor. With this new problem formulation, we propose two new approaches for training the emulator to match the attractor statistics.
>
> **Q4**:
> > There are other ways (e.g., MMD) to measure the distance of a distribution… It would be good to clarify the motivation…
>
> There are many ways of measuring differences between distributions, including point-wise divergences such as KL, moment-based distances such as MMD, as well as the Sinkhorn distance from optimal transport—which we use in the paper. Using MMD would result in a similar method to our OT approach and act as a drop-in replacement for the Sinkhorn distance for matching the distributions of summary statistics. In fact, we have included a new experiment where we train an emulator using a Gaussian-kernel MMD loss (Table R4). We find that MMD, even after careful hyperparameter tuning, does not perform as well as the Sinkhorn loss.
>
> We chose the Sinkhorn loss because it has many well-known advantages over point-wise divergences and kernel MMD distances: e.g., for two distributions that do not have overlapping support, the Wasserstein distance underlying the Sinkhorn loss still provides a high-quality gradient signal that makes it useful for optimization. MMD also requires many hyperparameter choices, including the choice of kernel and kernel-specific parameters, that have a significant effect on the properties of the final distance measure.
>
> Contrastive learning, on the other hand, allows us to learn a set of invariant statistics from the data. This provides a method that does not rely on prior domain knowledge about the chaotic attractor and instead uses the multi-environment setting to automatically learn informative statistics. This motivation is further discussed in the introduction (Sec. 1), when we propose the method (Sec. 3.2), and in the discussion (Sec. 5).
>
> [R1] Johannes Brandstetter, Daniel E. Worrall, Max Welling, Message Passing Neural PDE Solvers, ICLR, 2022.
>
> **L1**:
> > It would be good to discuss in detail the advantages and disadvantages…
>
> Please see the discussion in the general rebuttal.

---

> > ### Comment · Reviewer_tYBd · 2023-08-18
> > **Reply**
> >
> > Thank you for your response. I now have a better understanding of this paper, and the experimental results using MMD are also useful. I agree that the idea of using OT and CL to predict chaos dynamics is new in itself. I will raise my score by 1.

---

> > > ### Author Response · Authors · 2023-08-20
> > > **Thank you for your reply!**
> > >
> > > Dear reviewer, we are sincerely thankful for your constructive feedback and appreciation of our work! We will ensure to include the experimental results using MMD, and discussions of our OT and CL approaches in the final paper.

---

### Official Review · Reviewer_DhyB · 2023-07-04

**Soundness:** 2 fair
**Presentation:** 2 fair
**Contribution:** 3 good
**Rating:** 6
**Confidence:** 4

**Summary:**

This paper proposes two methods to preserve invariant measures of chaotic systems in the multi-environment setting when training neural operators. Given some expert knowledge of the underlying dynamical systems, they propose a new optimal transport loss, which uses this knowledge to match the statistics. Without expert knowledge, they use a contrastive learning approach to learn invariant statistics, which are used to construct the feature loss to preserve these invariant statistics. These two methods show better performance to reproduce the invariant statistics of noisy chaotic systems.

**Strengths:**

1. Long-time prediction of chaotic systems is different, this paper proposed two methods to better represent chaotic systems using neural operators by preserving the invariant statistics of chaotic attractors.
2. The methods are robust to noise and chaos.

**Weaknesses:**

1. These two methods are aimed at different problem settings: the optimal transport-based approach needs prior knowledge, while the contrastive learning approach does not; the optimal transport-based approach seems not to depend on the multi-environment setting, whereas the contrastive learning approach does depend on this setting.  These two methods are relatively separate, and the relationship between them is not close.
2. The two methods are mixed together, so the structure of the paper does not look clear.
3. It seems that the optimal transport-based approach requires too many equation constraints. For example, in the two examples given in the paper, almost all of the terms in the underlying dynamical equations are used.
4. As for the contrastive learning approach, higher requirements are put forward for the diversity of data.
5. The introduction of the Noise Contrastive Estimation (InfoNCE) loss is not described clearly in the main text.


**Questions:**

1. Is it possible to provide the problem setting, limitations, and advantages of the two methods more clearly, and clearly explain the difference between the two methods, instead of introducing them together?
2. Whether the performance can be maintained if more limited prior knowledge is utilized for the optimal transport-based approach?
3. If all those equation constraints have to be used to ensure performance, should it be compared with methods such as physical informed neural operator that utilize similar information instead of the conventional neural operator?
4. As for the contrastive learning approach, what kind of diversity does the data need to meet to ensure better performance? Could you provide an intuitive explanation?


**Limitations:**

The authors have adequately addressed the limitations.

---

> ### Author Rebuttal · Authors · 2023-08-10
>
> Thank you for your feedback!
>
> **W1**:
> > These two methods are… relatively separate…
>
> While the two proposed methods use different machine learning tools, they are very much related by a shared goal and overall approach: to train emulators to capture long-term chaotic behavior by preserving the invariant measure of the attractor. In fact, both methods go about this goal in conceptually similar ways, by matching the statistics learned by the emulator to statistics from the data. The primary difference is how those statistics are identified, whether it is from prior domain knowledge or learned directly from the data. Our contributions consist of both identifying and formulating the shared problem (Section 2) and then proposing two approaches for solving this problem (Sections 3.1 and 3.2). The fact that contrastive learning (CL) can preserve chaotic attractors nearly as well as methods that use prior knowledge is remarkable.
>
> The multi-environment setting provides an avenue for identifying the relevant statistics for characterizing chaotic attractors. The CL approach uses multiple environments explicitly during pre-training, while the OT approach uses this information implicitly as a part of the prior knowledge used for choosing informative summary statistics. In other words, the best summary statistics for training an emulator are precisely those that can tell the difference between environments/attractors and therefore provide a signal to push the emulator to match the correct attractor.
>
> The final paper will better emphasize these points.
>
> **W2**:
> > The two methods are mixed together…
>
> The two methods are presented in separate sections: physics-informed OT (Section 3.1) and unsupervised CL (Section 3.2). Both represent approaches for training an emulator to preserve the invariant measure of the attractor—the problem formulated in Section 2.
>
> **W3**:
> > It seems that the optimal transport-based approach requires too many equation constraints…
>
> We agree! This was the motivation for developing the CL approach. The OT approach is precisely designed to use existing domain knowledge to choose informative summary statistics and train a better emulator. That said, we are interested in better understanding how the quality and quantity of the chosen statistics influence the trained emulator. To better study this effect, we have performed additional experiments (Table R2 in the rebuttal PDF) with a reduced set of summary statistics including using a minimally informative statistic to demonstrate when the OT approach begins to fail due to a poor choice of statistic.
>
> **W4**:
> > As for the contrastive learning approach, higher requirements… for the diversity of data.
>
> The multi-environment setting is a very natural setting for many scientific and engineering applications, where different measured trajectories often have varying parameters due to environmental noise or varying control inputs. It also presents a more challenging generalization problem than the single-environment setting. As with any unsupervised representation learning method, CL requires sufficient data diversity, which in this case comes directly from the multi-environment setting. Please also see our response below to Q4.
>
> **W5**:
> > The introduction of the Noise Contrastive Estimation (InfoNCE) loss is not described clearly in the main text.
>
> We introduce InfoNCE in Section 3.2 eq. 15, cite its origin as a contrastive learning loss, and explain the intuition for its use and its relevance to our problem.
>
> **Q1**:
> > Is it possible to provide the problem setting, limitations, and advantages…
>
> The two methods share a problem setting as presented in Section 2. Please see the general rebuttal for additional discussion comparing the approaches.
>
> **Q2**
> > Whether the performance can be maintained if more limited prior knowledge…
>
> In our updated results (Table R2), we show the effect of using a smaller set of summary statistics as well as using a minimally informative statistic for the OT approach. You can often get away with using a fairly limited set of summary statistics if the chosen statistics are highly informative. If there is a real lack of prior knowledge, then our CL approach offers a very compelling alternative.
>
> **Q3**
> > If all those equation constraints have to be used to ensure performance, should it be compared with methods such as physical informed neural operator…
>
> We believe the reviewer means statistics instead of constraints since we impose no constraints. The OT approach matches summary statistics of the model outputs to those of the data and never treats those statistics as terms in a governing equation. Another important distinction to make here is that methods such as the physics-informed neural operator assume a *known* PDE and then use a neural operator to fit the solution, which can be done even without training data. We are using the neural operator to learn a time evolution operator directly from data without knowing the PDE. (We agree that assuming we know what statistics to preserve with the OT method can be problematic, so we developed the CL method to address this.)
>
> **Q4**:
> > As for the contrastive learning approach, what kind of diversity does the data need to meet to ensure better performance?...
>
> Intuitively, CL uses the data diversity provided by the multi-environment setting to choose invariant statistics by identifying which statistics are informative for distinguishing between the various attractors seen in the data. Empirically, our experiments show that randomly varying just one or a few system parameters is enough to provide the necessary data diversity for CL. In our updated results (Table R3), we show a new experiment that further reduces the range of the only varying parameter F in the Lorenz-96 dataset from [10, 18] to [16, 18]. Even with such minimal diversity, we still see good performance using our CL approach.

---

> > ### Comment · Reviewer_DhyB · 2023-08-12
> >
> > Thank you for your detailed replies, which cleared up most of my confusion. The added experiments explain the performance of the model in the statistically limited case, making the discussion more systematic. Based on this, I increased the score accordingly.

---

> > > ### Author Response · Authors · 2023-08-12
> > > **Thank you for your reply!**
> > >
> > > Thank you for your constructive feedback of our paper and appreciation of our work! We will make sure to include the performance of the OT using limited statistics and other discussed clarifications, in the revision of our paper.

---

### Official Review · Reviewer_fKLU · 2023-07-04

**Soundness:** 2 fair
**Presentation:** 3 good
**Contribution:** 2 fair
**Rating:** 5
**Confidence:** 4

**Summary:**

The authors proposed a training framework to preserve preserve invariant measures of chaotic attractors. First, they identify training standard neural operators using MSE on chaotic dynamics does not work. Then they suggested to train neural operators to preserve the invariant measures of chaotic dynamics and the time-invariant statistics. Importantly, they proposed two approaches to train neural operators to preserve those statistics. One is the optimal transport-based approach, and the other is the contrastive learning approach. They  empirically showed that both approaches capture the true invariant statistics.

**Strengths:**

The paper is well written and organized. The paper follows a well-defined structure, with a logical flow of ideas throughout.

**Weaknesses:**

The experiment part could benefit from more substantial content. Comparison with previous methods should be addressed in the experiment part. See below for the two questions as well.

**Questions:**

What is the advantage of preserving the invariant features in chaotic dynamics? Does it really contribute to long-term forcasting, which suppose to be the objective? If it does, I expect the author to show an improvement in long-term forcasting quatitatively by adopting one of the two proposed approches.

The proposed method is evaluated on the synthetic data, the Lorenz-96 and the Kuramoto–Sivashinsky chaotic dynamics. I am wondering how the model performs on some real world data, e..g, the climate system.

**Limitations:**

The authors have addressed the limitations

---

> ### Author Rebuttal · Authors · 2023-08-10
>
> Thank you for your feedback!
>
> **W1**:
> > The experiment part could benefit from more substantial content. Comparison with previous methods should be addressed in the experiment part. See below for the two questions as well.
>
> Prior methods for training emulators generally use the standard MSE loss which we show as the baseline in our results. We also include a comparison with a previous approach based on the Sobolev norm (Table 2). Our approaches outperform the Sobolev norm method in our noisy chaotic settings.
>
> **Q1.1**:
> > What is the advantage of preserving the invariant features in chaotic dynamics?
>
> Preserving invariant features of chaotic dynamics means matching the long-term statistical behavior of a chaotic dynamical system. For example, weather models may only be able to predict the exact weather up to two weeks ahead of time. In fact, it is impossible to predict the precise state of the Earth on the time scales (years) required for climate modeling [1], but we would still like to predict important statistical features of the climate, e.g. the average number of hurricanes per year. It is these kinds of statistical features of the chaotic dynamics that we want to preserve for long-term forecasting.
>
> **Q1.2**:
> > Does it really contribute to long-term forecasting, which suppose to be the objective?
>
> Yes! Any long-term prediction for the exact state of the system diverges exponentially from the ground truth due to chaos, so the important quantities to consider for long-term forecasting are precisely the statistical features of the dynamics. By matching these statistical features, we are able to train an emulator that truly captures the chaotic dynamics and correctly models the only predictable aspects of the chaotic system.
>
> Another important point here is that short-term prediction performance does not necessarily imply long-term performance in terms of statistics and model stability. An accurate weather model run for a longer time will not necessarily produce high-quality climate statistics, just as an emulator trained only on short-term MSE may fail to capture the true long-term behavior of the system. The goal of our work is to help rectify this problem by proposing methods that directly tackle long-term forecasting for chaotic systems.
>
> **Q1.3**:
> > If it does, I expect the author to show an improvement in long-term forecasting quantitatively by adopting one of the two proposed approaches.
>
> Our results *do* show a significant improvement (which is often even visually evident in the sample predictions) over the standard MSE-based training method on the relevant long-term statistical behavior. In particular, our metrics include the distributions of key summary statistics and the energy spectrum, both of which represent long-term statistical characterizations of the chaotic system. Our trained deterministic emulators provide much higher quality instances of the system dynamics, which sample the chaotic attractor, and this is the best you can hope for when performing long-term forecasts of chaotic dynamics. The new experiments (see general rebuttal) provide additional evidence for this using new metrics.
>
> The final paper will better emphasize these points.
>
> **Q2**:
> > The proposed method is evaluated on the synthetic data, the Lorenz-96 and the Kuramoto–Sivashinsky chaotic dynamics. I am wondering how the model performs on some real world data, e..g, the climate system.
>
> We are also excited to see how our approaches would scale for climate modeling. However, this is outside the scope of this work. Existing climate emulators, such as FourCastNet [2] and ClimaX [3], are large models that cost millions to train, tune, and validate.
>
> [1] What might we learn from climate forecasts? Smith, Leonard A. (2002).
>
> [2] FourCastNet: A Global Data-driven High-resolution Weather Model using Adaptive Fourier Neural Operators. Pathak, Jaideep et al. (2022).
>
> [3] ClimaX: A foundation model for weather and climate. Nguyen, Tung et al. (2023).

---

> > ### Comment · Reviewer_fKLU · 2023-08-16
> > **Thank you for answering my questions**
> >
> > Re Q1.1: I think there is still a large gap between 'the average number of hurricanes per year' and predicting statistical features of the chaotic dynamics...To me, this model still cannot tell me the exact long-term prediction of chaotic dynamics. Some of the statistical features can be visualized by for example, PCA, t-SNE based on history data. Why do we need to use the model to tell us these low-dimensional invariant featuers?
> >
> > Re Q1.2: Can the authors give an example of the invariant feature of a REAL chaotic system? The Lorenz-96 is a synthetic system with a known low-dimensional feature. If the invariant feature of a real chaotic system is too complex, or too hard to define, how to verify the model can be applied to a real system?
> >
> > Generally I kind of agree with Reviewer GF1X that '...to see how the methods perform on at least one empirical chaotic problem would have also been nice...'

---

> > > ### Author Response · Authors · 2023-08-17
> > > **Response to your questions (1/2)**
> > >
> > > Dear reviewer, we greatly appreciate your prompt feedback. Please allow us to address the concerns you've highlighted.
> > >
> > > > Re Q1.1: I think there is still a large gap between 'the average number of hurricanes per year' and predicting statistical features of the chaotic dynamics…
> > >
> > > The “average number of hurricanes” is an example of a statistical feature of climate models. Of course, there are many other important statistical features that will be relevant for building a high-quality climate emulator. Our goal is to use these statistical features to help train better deterministic emulators that will reproduce the true underlying chaotic attractor. Moreover, our CL approach is precisely focused on automatically identifying a rich collection of informative statistical features without requiring prior knowledge. Our results show that training an emulator using the invariant statistics learned from CL significantly improves the ability of the emulator to preserve many physics-informed statistical metrics we use for evaluation.
> > >
> > >
> > > > To me, this model still cannot tell me the exact long-term prediction of chaotic dynamics.
> > >
> > >
> > > No model can even *in theory* tell us the exact long-term prediction of chaotic dynamics. This is because chaotic systems exhibit extreme sensitivity to initial conditions, i.e., any small change or noise in the initial condition results in an exponentially divergent solution, even if we use a mathematically exact model. That is why we focus on statistical features that characterize chaotic dynamics. During the evaluation, we use statistics-based metrics that characterize the chaotic attractor of our simulated systems and show that these indeed match the data well when performing long-term predictions: 1,000 time steps in the Lorenz 96 system and 500 time steps in the Kuramoto–Sivashinsky system.
> > >
> > >
> > > Moreover, lots of studies related to climate science have shown that preserving the statistical features of chaotic systems is very important in predicting and understanding the climate. Fundamental studies ([1, 2]) show that the dimension of attractors characterizes the minimum number of variables to describe the system. [3, 4] also demonstrate that preserving the invariant attractors is helpful for determining extreme events. A recent study ([4]) also identified rMSE as a problematic evaluation metric for climate prediction as it implicitly assumes two time series should be aligned for every individual day, which is nearly impossible for real data. Instead, they use an invariant summary statistic for evaluating climate models.
> > > > Some of the statistical features can be visualized by for example, PCA, t-SNE based on history data. Why do we need to use the model to tell us these low-dimensional invariant featuers?
> > >
> > >
> > > Yes, PCA, or t-SNE can be used to visualize the features. However, these features do not come for free, and our goal is not visualization. While PCA or t-SNE can visualize known features, these methods do not allow us to discover new invariant statistical features. We also do not specifically target low-dimensional or visualizable features. In fact, our CL approach allows us to learn general high-dimensional invariant statistics without prior knowledge, which can then be used for training emulators.

---

> > > > ### Author Response · Authors · 2023-08-17
> > > > **Response to your questions (2/2)**
> > > >
> > > > > Re Q1.2: Can the authors give an example of the invariant feature of a REAL chaotic system?
> > > >
> > > >
> > > > The four features we adopted in the evaluation of our experiments have been widely used to characterize the chaotic systems in real applications, e.g. climate:
> > > >
> > > > 1. The joint distribution of physics-informed summary statistics is a common choice for evaluating the quality of emulators for complex chaotic systems ([5]).
> > > >
> > > > 2. The Fourier energy spectrum, which described the distribution of kinetic energy as a function of frequency, is a common high-dimensional invariant statistic used in the analysis of fluid systems ([6, 7]).
> > > >
> > > > 3. The leading Lyapunov exponent (LLE) is a widely adopted measurement that characterizes how fast the dynamical states diverge with respect to time ([2, 8]).
> > > >
> > > > 4. The fractal dimension ([1, 9]), as a measure of complexity, has been used to describe the spatiotemporal climate data.
> > > >
> > > >
> > > > > The Lorenz-96 is a synthetic system with a known low-dimensional feature. If the invariant feature of a real chaotic system is too complex, or too hard to define, how to verify the model can be applied to a real system?
> > > >
> > > >
> > > > Our proposed CL approach is exactly designed to deal with complex systems where there is no significant prior knowledge! While simulated, Lorenz-96 and Kuramoto–Sivashinsky represent *high-dimensional* chaotic attractors with origins as models for fluid turbulence, and our experiments show that our proposed approaches work well for training better emulators for such systems. We are not aware of a low-dimensional characterization of the Lorenz-96 or Kuramoto–Sivashinsky attractor.
> > > >
> > > >
> > > > > Generally I kind of agree with Reviewer GF1X that '...to see how the methods perform on at least one empirical chaotic problem would have also been nice...'
> > > >
> > > >
> > > > We would also love to evaluate on empirical data! However, as we discussed in our response to reviewer GF1X, there is very little publicly available empirical data for systems as complex as the spatiotemporal chaos shown in our experiments.
> > > > We would love to see this change in the future. However, current spatiotemporal modeling papers are almost all evaluated on high-quality simulations. The one exception is weather data used for training emulators such as FourCastNet [10] and ClimaX [11], which are very large models that cost millions to train. We would love to see how well these approaches scale to such models but that is outside the scope of this work.
> > > >
> > > >
> > > > Thank you again for your time providing the feedback! If there are any further questions or details you’d like to discuss, we are here to assist. We look forward to hearing back from you.
> > > >
> > > >
> > > > [1] Estimating the Dimensions of Weather and Climate Attractors. Fraedrich, Klaus. (1986)
> > > >
> > > > [2] Predicting uncertainty in forecasts of weather and climate, Palmer, T N. (2000)
> > > >
> > > > [3] Universal behavior of extreme value statistics for selected observables of dynamical systems. Lucarini, Valerio, et al. (2011)
> > > >
> > > > [4] A locally time-invariant metric for climate model ensemble predictions of extreme risk. Virdee, Mala, et al. (2023)
> > > >
> > > > [5] Multiscale Simulations of Complex Systems by Learning their Effective Dynamics. Vlachas, Pantelis R., et  al. (2021)
> > > >
> > > > [6] Characterization and prediction of runoff dynamics: a nonlinear dynamical view. Islam, M.N, Sivakumar, B. (2002)
> > > >
> > > > [7] Global energy spectrum of the general oceanic circulation. Storer, Benjamin A., et al. (2022)
> > > >
> > > > [8] Predictability of Weather and Climate. Krishnamurthy, V. (2019)
> > > >
> > > > [9] Estimating the Fractal Dimension and the Predictability of the Atmosphere. Zeng, X., et al. (1922)
> > > >
> > > > [10] FourCastNet: A Global Data-driven High-resolution Weather Model using Adaptive Fourier Neural Operators. Pathak, Jaideep et al. (2022)
> > > >
> > > > [11] ClimaX: A foundation model for weather and climate. Nguyen, Tung et al. (2023)

---

### Official Review · Reviewer_G25b · 2023-07-15

**Soundness:** 3 good
**Presentation:** 4 excellent
**Contribution:** 3 good
**Rating:** 6
**Confidence:** 4

**Summary:**

The paper uses neural operators to track the invariant statistics of chaotic systems. It novelly proposes to use the optimal transport loss and contractive learning to match the distribution, so that the learned model correctly track the attractor among the chaotic behavior. The paper uses FNO as a backbone to study the Lorenz and KS equation. Experiments show the proposed methods better track the behaviors such as histogram and spectrum.

**Strengths:**

The paper proposes to learn the dynamics of chaotic system with the optimal transport loss and contractive learning. The proposed methods are more stable compared to the standard MSE/L2 loss, and they can capture the attractor of the system. Experiments show these methods better track the statistical behaviors such as histogram and spectrum.

**Weaknesses:**

The paper only studied Lorenz and 1d Kuramoto–Sivashinsky equation. It will be very interesting to study how these methods scale to 2d Navier-Stokes (Kolmogorov Flow) problem.

It will be interesting to discuss time complexity.

**Questions:**

I have a few questions:
- Which method (Sinkhore algorithm and contrastive learning) does better? What are their comparative advantages and disadvantages?
- How does the training time of the Sinkhore algorithm and contrastive learning? I assume they will be slower than the standard supervised learning. It will be great to report the runtime and training time. Would the wasserstein loss be easier to optimize compared to L2 loss? It will be interesting to show a training curve.
- How does the proposed methods compared to generative model [1, 2]. For example, in Generative adversarial neural operators they also matches the wasserstein distance.

[1] Rahman, Md Ashiqur, et al. "Generative adversarial neural operators." arXiv preprint arXiv:2205.03017 (2022).

[2] Lim, Jae Hyun, et al. "Score-based diffusion models in function space." arXiv preprint arXiv:2302.07400 (2023).

**Limitations:**

The paper addressed the limitation.

---

> ### Author Rebuttal · Authors · 2023-08-10
>
> Thank you for your feedback!
>
> **W1**:
> > It will be very interesting to… scale to 2d Navier-Stokes…
>
> We agree that scaling to 2D Navier–Stokes and even higher-dimensional chaotic problems would be an interesting extension to this work. We note, however, that our current results on 1D spatiotemporal chaos already provide high-dimensional chaotic attractors that allow us to validate our approaches and show the benefits of using chosen or learned statistics to train modern deep learning-based emulators. There is no fundamental impediment to extending our method to this setting; it simply requires using a very different backbone in our architecture and more training time than the rebuttal period allows.
>
> **W2**:
> > It will be interesting to discuss time complexity.
>
> In the rebuttal PDF Figure R1, we document the training time of the models. The OT approach relies on the Sinkhorn algorithm which scales as $O(n^2\log(n))$ for comparing two distributions of $n$ points each (Theorem 2 in [3]). In our experiments, we use $n = 6000$ to $n = 25,600$ points with no issues, so this approach scales relatively well. The CL approach requires pretraining but is even faster during emulator training since it uses a fixed, pre-trained embedding network.
>
> We will gladly include a more detailed discussion of time complexity in our final paper.
>
> **Q1**:
> > Which method (Sinkhore algorithm and contrastive learning) does better? What are their comparative advantages and disadvantages?
>
> Please see the general rebuttal for a detailed discussion of this topic. We will make sure to include this discussion in our final paper.
>
> **Q2**:
> > How does the training time of the Sinkhore algorithm and contrastive learning? I assume they will be slower than the standard supervised learning. It will be great to report the runtime and training time.
>
> We show training times in the rebuttal PDF Figure R1. CL is faster than Sinkhorn (OT) during emulator training because CL uses a pre-trained embedding function. However, CL does require a pre-training step.
>
> > Would the wasserstein loss be easier to optimize compared to L2 loss? It will be interesting to show a training curve.
>
> The Sinkhorn loss is a computationally-efficient proxy for the Wasserstein distance. While the Sinkhorn loss is still slower to compute than the L2 loss (rMSE), it appears to converge faster. We include sample training curves in the rebuttal PDF showing that the Sinkhorn loss converges in fewer iterations than rMSE.
>
> **Q3**:
> > How does the proposed methods compared to generative model [1, 2]. For example, in Generative adversarial neural operators they also matches the wasserstein distance.
>
> Both cited works are generative models, as you point out. However, our problem is not a generative modeling problem. We are not trying to directly sample from the attractor distribution but instead find and use high quality attractor statistics that allow us to train a better emulator for the dynamical system. As the emulator evolves over time, it may be effectively sampling from the attractor distribution, but the sampling itself is not the end goal. Replacing the emulator with a general-purpose generative model does not accomplish our goal of training a model that accurately captures the time dynamics of the system.
>
> To address the use of the Wasserstein distance in [1], the generative adversarial neural operator (GANO) uses the Wasserstein distance in the same way as the original Wasserstein GAN uses the Wasserstein distance—as a motivation for the adversarial framework for generative modeling. As noted in [1], “while the cost functional… is well defined, showing that the learned measure is indeed an approximation of [the true distribution]... remains an open problem. We address this issue empirically and perform a set of experiments that demonstrate that GANO produces diverse outputs from the data probability measure.” In other words, it is not clear whether GANs really do compute a Wasserstein approximation and we should really treat the connection to OT as general motivation. In our work, we use the Sinkhorn algorithm to solve an entropy-regularized OT problem, which is a very well-understood approximation theoretically, and then use the Sinkhorn-approximated Wasserstein distance to train an emulator for a dynamical system rather than perform generative modeling.
>
> [1] Rahman, Md Ashiqur, et al. "Generative adversarial neural operators." (2022).
>
> [2] Lim, Jae Hyun, et al. "Score-based diffusion models in function space." arXiv preprint (2023).
>
> [3] Dvurechensky, Pavel, et al. “Computational Optimal Transport: Complexity by Accelerated Gradient
> Descent Is Better Than by Sinkhorn’s Algorithm.” (2018).

---

### Official Review · Reviewer_GF1X · 2023-07-24

**Soundness:** 2 fair
**Presentation:** 3 good
**Contribution:** 4 excellent
**Rating:** 7
**Confidence:** 4

**Summary:**

This work considers the problem of learning invariant statistics of chaotic dynamical systems. It is motivated by the observation that training neural operators by standard MSE loss focuses on short-term predictability and thus may miss an attractor’s properties. Two ways of augmenting the MSE loss for capturing invariant stats are discussed: One based on the Wasserstein distance between prob distributions across explicitly provided stats (physics-informed), and one based on contrastive feature learning. Experiments on chaotic benchmarks, the Lorenz-96 and the Kuramoto-Sivashinsky equations, are shown.

**Strengths:**

The paper is timely and addresses an important issue, namely how we can capture crucial properties of an observed dynamical system beyond just short-term predictions. In general I like the approach, and think this could be a very fruitful addition to the current state. In general, the paper is also well written, and the ideas well motivated. Although incorporating invariant stats into the loss has recently been discussed, this paper offers a different perspective on this topic.

**Weaknesses:**

On the other hand, the experimental evaluations are, in my view, quite weak, and also a lot of related and relevant literature is not discussed.

The experiments primarily show that on any of the evaluation measures that particular method wins, for which exactly that measure has been included or accentuated in the loss function: If the loss is only MSE, the only-MSE trained method wins on short-term predictions; if the loss includes invariant statistics, the method wins on exactly those invariant stats. This is not surprising, but would be expected for almost anything explicitly included in the loss term (i.e., the method for which a specific property was included in the loss should have an edge when evaluated on that particular property).

It would have been much more convincing if methods which include invariant statistics could also reproduce many *other* properties of the attractor that were *not* explicitly included in the loss, like its Lyapunov spectrum or fractal dimension.
Besides, two of the columns in Table 1 \& 2 don’t contain any indication of statistical error and are therefore somewhat meaningless in my mind (are the differences significant or negligible?).
To see how the methods perform on at least one empirical chaotic problem would have also been nice.

Literature-wise, the idea of including invariant statistics in the loss has been considered by a number of recent publications, for instance https://arxiv.org/pdf/2304.12865.pdf or https://pubs.aip.org/aip/cha/article-abstract/33/6/063152/2900453/Learning-dynamics-on-invariant-measures-using-PDE?redirectedFrom=fulltext.
The fact that purely MSE-based methods have difficulty capturing invariant stats for chaotic systems has recently also been extensively discussed in https://proceedings.neurips.cc/paper_files/paper/2022/hash/495e55f361708bedbab5d81f92048dcd-Abstract-Conference.html and, relatedly, https://arxiv.org/abs/2306.04406, although these authors arrive at different conclusions of how to tackle the problem.
My perception is that this problem has received a lot more attention recently than the authors’ related works section makes one believe.

**Questions:**

My primary concern is really the weaknesses in the experimental evaluations as listed above. This sect. was a bit disappointing to read after the exciting start of the paper and presentation of methods.

**Limitations:**

Yes.

---

> ### Author Rebuttal · Authors · 2023-08-10
>
> Thank you for your feedback!
>
> **W1**:
> > The experiments primarily show that on any of the evaluation measures that particular method wins…
>
> We propose emulators with much more consistent long-term behavior and that accurately capture the dynamics of the chaotic system. We make significant gains in long-term behavior, as measured by the evaluation statistics, while retaining very similar short-term prediction performance. Both methods (OT and CL) achieve these benefits, with the contrastive method requiring no prior knowledge and being faster.
>
> For the CL method, we do not provide any explicit statistics during training. Instead, the contrastive pre-training automatically learns useful invariant statistics. For the OT approach, we show that training using a chosen set of informative summary statistics improves performance on those statistics as well as other statistics (e.g. the energy spectrum, Lyapunov exponent) not used during training.
>
> **W2**:
> > It would have been much more convincing if methods… reproduce… Lyapunov spectrum or fractal dimension.
>
> We have included estimates of the leading Lyapunov exponent and the fractal dimension in our updated results (Table R1). Note that our method does not aim to regularize dynamical features, such as the Lyapunov exponents, which are distinct from time-invariant statistics. Despite this, we see significant improvements to the model’s leading Lyapunov exponent in high-noise settings. Since our data consists of high-dimensional chaotic attractors, it is very difficult to obtain reliable estimates of the full Lyapunov spectrum and the fractal dimension [1]. However, we do see evidence that the estimated fractal dimension of our CL approach is better than the baseline.
>
> **W3**:
> > Besides, two of the columns in Table 1 & 2 don’t contain any indication of statistical error…
>
> Thank you for pointing this out. We have now included quantile intervals in our updated tables. Our results show significantly better performance on long-term statistics while retaining similar short-term prediction performance.
>
> **W4**:
> > To see how the methods perform on at least one empirical chaotic problem would have also been nice.
>
> Our work, like almost all others in this area, uses simulated data for validation. The datasets we currently use are representative of high-dimensional spatiotemporal chaos. It is quite difficult to find publicly available empirical datasets of fully-observed chaotic dynamical systems, especially ones showing spatiotemporal chaos. This is why most papers on spatiotemporal emulators are evaluated on simulated data. One publicly available dataset is global weather data, which we are excited to try but it would require significant resources.
>
> **W5**:
> > Literature-wise, the idea of including invariant statistics in the loss has been considered by a number of recent publications…
>
> We first note that 3 of the 4 papers ([1], [2], and [4]) should be considered contemporaneous work under the NeurIPS guidelines. That said, we would be happy to cite and add a discussion of all four papers to better clarify our work in light of very recent developments in the field.
>
> Two of the suggested papers ([3], [4]) focus on training traditional RNNs to emulate chaotic dynamics. Their methods and the ones presented in our submission have similar training protocols for the rMSE loss. Key differences between these papers and our approaches are (a) they only work in the noise-free setting whereas we assume noisy observations, (b) they only use a short-term rMSE loss, and (c) they evaluate their method in the easier single-environment setting. In our experiments, we see that rMSE is a fine loss function in noise-free settings (consistent with the conclusions of [3],[4]), but this loss fails to preserve chaotic attractors in noisy settings.
>
> [1] focuses on reservoir computing—an RNN-like architecture with a single trained output layer—and trains using known dynamical invariants, e.g. Lyapunov exponents. Training using the Lyapunov exponents (a) requires significant prior physical knowledge or (b) empirical estimates of the Lyapunov exponents, which are difficult to estimate stably from data, particularly in the presence of noise and high dimensions. Our updated evaluation results (Table R1) show that our training methods result in accurate estimates of the Lyapunov exponent even when it’s not used during training.
>
> The method for modeling chaotic systems suggested in [2] requires solving a PDE for the probability density of the state and differs significantly from our proposed approaches, which are based on chosen or learned statistics. In particular, [2] scales poorly to higher dimensional state spaces, like the spatiotemporal dynamics that we consider, because even explicitly representing a high-dimensional probability distribution—let alone solving a high-dimensional PDE using a mesh—suffers from the curse of dimensionality. This is likely why the experiments in [2] focus on low-dimensional (up to 3) dynamical systems since solving a 3D PDE (i.e., a PDE with three “channels” like Lorenz-63) can already be numerically challenging. In contrast, our experiments are on high-dimensional spatiotemporal systems with state space dimensions ranging from 60 to 256.
>
> [1] Constraining Chaos. Platt et al, April 2023.
>
> [2] Learning dynamics on invariant measures using PDE-constrained optimization. Botvinick-Greenhouse et al, Chaos, June 2023.
>
> [3] On the difficulty of learning chaotic dynamics with RNNs. Mikhaeil et al, NeurIPS 2022.
>
> [4] Generalized Teacher Forcing for Learning Chaotic Dynamics. Hess et al, June 2023.
>
> **Q1**:
> > My primary concern is really the weaknesses in the experimental evaluations as listed above.
>
> Please see the global rebuttal. We hope that our additional evaluation metrics and the above discussion address your concerns!

---

> > ### Comment · Reviewer_GF1X · 2023-08-18
> > **remaining points**
> >
> > I thank the authors for their response which clarified some of my points.
> >
> > A few remaining remarks:
> > - W1: This doesn't really address my point, I think. My argument was that it's quite expected that if you optimize method 1 for property A, then method 1 will outperform others not optimized for A. The additional evaluations on LE and FD mostly address this point, however.
> > - W2: Since this is all model-based, you have the Jacobians, so why can't you assess the whole LE spectrum?
> > - W4 \& W5: The authors state as a key difference to [4,5] that these do not involve settings with noise. Didn't check this, but what I do recall is that [4,5] used several *empirical* datasets which I would assume contain a lot of noise! So this falsifies the authors' statements about both the noise and the unavailability of empirical datasets with characteristics of chaos.

---

> > > ### Author Response · Authors · 2023-08-19
> > > **Response to your questions (1/2)**
> > >
> > > Dear reviewer, we greatly appreciate your valuable feedback! Please allow us to address your comments as follows.
> > >
> > > > W1: This doesn't really address my point, I think. My argument was that it's quite expected that if you optimize method 1 for property A, then method 1 will outperform others not optimized for A. The additional evaluations on LE and FD mostly address this point, however.
> > >
> > > We agree with this point. To some extent, the OT method can be thought of as an “oracle” approach that lets us measure a best-case level of performance that we would aim for with a method that cannot be optimized for property A. This is why we chose to include performance on property A among our results. But the results are certainly strengthened with your suggestions of LE and FD.
> > >
> > >
> > > It is also important to note that our OT method also performs significantly better than the baseline when we evaluated the performance using Fourier energy spectrum, and the leading Lyapunov exponent in the high noise setting.
> > >
> > >
> > > In addition, new experiments with our OT method (also mentioned in our discussion with Reviewer DhyB) demonstrate that OT could also improve performance when using limited knowledge of the invariant statistics, which implies the better generalization ability of the OT approach.
> > >
> > >
> > > Last, we want to emphasize our CL approach, not designed explicitly on any evaluation metrics (with the absence of physics-informed prior knowledge), delivers superior performance than the rMSE baseline on all four metrics we evaluated on.
> > > > W2: Since this is all model-based, you have the Jacobians, so why can't you assess the whole LE spectrum?
> > >
> > >
> > > Thank you for your suggestion! We initially focused on obtaining LLEs since it is a much more straightforward calculation, but we have now obtained estimates for the full Lyapunov spectrum, as requested. We present the results regarding the Lyapunov spectrum in the table below. For the Lyapunov Spectrum Error, we report the sum of relative absolute errors across the full spectrum: $\sum_i^d |\hat{\lambda}_i - \lambda_i| / |\lambda_i|$, where $d$ is the dimension of the dynamical state (i.e., 60 for Lorenz 96). As suggested by [5], we also compare the number of positive Lyapunov exponents (LEs) as an additional statistic to measure the complexity of the chaotic dynamics. We compute the absolute error in the number of positive LEs $|\sum_i^d \mathbf{1}(\hat{\lambda}_i > 0) - \sum_i^d \mathbf{1}(\lambda_i > 0)|$ averaged over the test instances.
> > >
> > > |r| Training | Leading LE Error $\downarrow$ | Lyapunov Spectrum Error $\downarrow$| Total number of positive LEs Error $\downarrow$|
> > > |-------------|-------------|-------------|-------------|-------------|
> > > |0.1 |$\ell_{\rm rMSE}$ | **0.014** (0.006, 0.021) | 0.388 (0.110, 0.309)  | 0.526 (0.000, 1.000) |
> > > |0.1 | $\ell_{\rm sinkhorn} + \ell_{\rm rMSE} $|  0.049 (0.040, 0.059) | **0.256** (0.168, 0.285) | 0.375 (0.000, 1.000) |
> > > |0.1 | $\ell_{\rm feature}  + \ell_{\rm rMSE} $| 0.065 (0.058, 0.073) |0.285 (0.164, 0.289) | **0.365** (0.000, 1.000) |
> > > ||||||
> > > |0.2 | $\ell_{\rm rMSE}$ | 0.175 (0.156, 0.191) | 1.940 (0.522, 0.726) | 4.248 (4.000, 5.000) |
> > > |0.2 |$\ell_{\rm sinkhorn} + \ell_{\rm rMSE} $|  0.019 (0.006, 0.030) | 0.837 (0.122, 0.590) | **2.540** (2.000, 3.000) |
> > > |0.2 |$\ell_{\rm feature}  + \ell_{\rm rMSE} $| **0.012** (0.006, 0.018) | **0.769** (0.138, 0.568) | 2.819 (2.000, 3.000) |
> > > ||||||
> > > |0.3 |$\ell_{\rm rMSE}$ |  0.446 (0.425, 0.463) | 1.979 (0.702, 0.939) | 7.230 (7.000, 8.000) |
> > > |0.3 |$\ell_{\rm sinkhorn} + \ell_{\rm rMSE} $|  0.101 (0.062, 0.134) | **1.186** (0.571, 0.745) | **4.824** (4.000, 6.000) |
> > > |0.3 |$\ell_{\rm feature}  + \ell_{\rm rMSE} $| **0.067** (0.045, 0.091) | 1.290 (0.558, 0.780) | 5.603 (5.000, 6.000) |
> > >
> > > Table 1. **Performance with varying noise level on Lorenz-96 evaluated on Lyapunov spectrum.** The average (25th, 75th percentile) error rates over 200 testing instances of training the neural operator with (1) only rMSE; (2) Sinkhorn (OT) and rMSE (using prior knowledge to choose summary statistics); and (3) contrastive feature loss (CL) and rMSE (no prior knowledge used). In the presence of high noise, OT and CL give lower relative errors on the Leading Lyapunov exponent (LLE). When evaluating the full Lyapunov spectrum, OT and CL show significant advantages than the baseline. In addition, the lower absolute errors of the total number of the positive Lyapunov exponents (LEs) suggest that OT and CL are able to match the complexity of the true chaotic dynamics.

---

> > > > ### Author Response · Authors · 2023-08-19
> > > > **Response to your questions (2/2)**
> > > >
> > > > > W4 & W5: The authors state as a key difference to [4,5] that these do not involve settings with noise. Didn't check this, but what I do recall is that [4,5] used several empirical datasets which I would assume contain a lot of noise! So this falsifies the authors' statements about both the noise and the unavailability of empirical datasets with characteristics of chaos.
> > > >
> > > >
> > > > We thank you for your further questions! As there is no [5] in the original reference list, we assume you are pointing to [3] and [4].
> > > >
> > > >
> > > > Our work focuses on training neural operators, like the Fourier neural operator (FNO), on fully observed high-dimensional deterministic chaos, usually in the form of spatiotemporal systems governed by PDEs, and not on general time-series data. The empirical datasets, such as the ECG and EEG datasets ([4]), are very much partially observed and potentially stochastic and so do not fall under the problem statement we describe in Section 2. Additionally, we should note that the empirical datasets in [3] and [4] are single-environment problems and cannot be used to test our multi-environment approaches. We would ultimately like to apply this approach to train emulators using empirical fluid dynamics data or global weather data, like the data used to train FourCastNet ([6]) and ClimaX ([7]), where the goal is to predict spatiotemporal climate dynamics for multiple subregions and time periods. We will make sure to clarify this in the final paper.
> > > >
> > > >
> > > > Regarding the noise setup, you are right! [3] and [4] do test on empirical time-series data which is expected to contain noise. [3] states in their empirical study that they operate on de-noised and smoothed data after preprocessing the raw data ([8]), while [4] does not apply preprocessing. We will amend our discussion to reflect this in our final paper. However, on their simulated datasets, which are closer to our problem setup, [3] does not add any noise and [4] only adds up to 5% ($r=0.05$) noise, which is much lower than the noise in our experiments.
> > > >
> > > >
> > > > Last, we want to emphasize that both [3] and [4] are only using the short-term rMSE loss, which is our baseline in the experiments, and perform poorly in chaotic setups.
> > > >
> > > >
> > > > We thank you again for your time providing us with the feedback! We are happy to assist if there are any questions or details you would like to discuss. We are looking forward to hearing back from you.
> > > >
> > > >
> > > > [1] Constraining Chaos. Platt et al, April 2023.
> > > >
> > > > [2] Learning dynamics on invariant measures using PDE-constrained optimization. Botvinick-Greenhouse et al, Chaos, June 2023.
> > > >
> > > > [3] On the difficulty of learning chaotic dynamics with RNNs. Mikhaeil et al, NeurIPS 2022.
> > > >
> > > > [4] Generalized Teacher Forcing for Learning Chaotic Dynamics. Hess et al, June 2023.
> > > >
> > > > [5] Constructing Discrete Chaotic Systems with Positive Lyapunov Exponents. Wang, Chuanfu, et al. (2017)
> > > >
> > > > [6] ClimaX: A foundation model for weather and climate. Nguyen, Tung et al. (2023)
> > > >
> > > > [7] FourCastNet: A Global Data-driven High-resolution Weather Model using Adaptive Fourier Neural Operators. Pathak, Jaideep et al. (2022)
> > > >
> > > > [8] Nonlinear noise reduction: A case study on experimental data.  Kantz, H., et al. (1993)

---

> > > > > ### Comment · Reviewer_GF1X · 2023-08-19
> > > > > **one final point**
> > > > >
> > > > > I thank the authors very much for their thorough response.
> > > > > There is one last point that I’d like to see clarified though: The authors state that training using just rMSE will not do a good job on chaotic attractors. However, I brought up [3,4] precisely because in those papers only rMSE is used (plus some amendments to the training procedure), yet apparently superb recovering of chaotic long-term behavior is achieved. Thus, that these approaches perform poorly in chaotic setups is not true, it's in my understanding actually what they were designed for!
> > > > > I do think that the authors’ idea of loss criteria including long-term statistics is an important conceptual contribution, but apparently this is not the only route for recovering long-term statistics, even for highly chaotic systems; similar for [1,2] who augment the loss through LE/FD. A fair and balanced discussion should probably make that clear.

---

> > > > > > ### Author Response · Authors · 2023-08-20
> > > > > > **Response to your questions.**
> > > > > >
> > > > > > Dear reviewer, we thank you for your prompt feedback! Once again, we are sincerely thankful for your suggestions on evaluating the Lyapunov spectrum and fractal dimension. Please allow us to address your concerns for references as follows.
> > > > > >
> > > > > > > I thank the authors very much for their thorough response. There is one last point that I’d like to see clarified though: The authors state that training using just rMSE will not do a good job on chaotic attractors. However, I brought up [3,4] precisely because in those papers only rMSE is used (plus some amendments to the training procedure), yet apparently superb recovering of chaotic long-term behavior is achieved. Thus, that these approaches perform poorly in chaotic setups is not true, it's in my understanding actually what they were designed for! I do think that the authors’ idea of loss criteria including long-term statistics is an important conceptual contribution, but apparently this is not the only route for recovering long-term statistics, even for highly chaotic systems; similar for [1,2] who augment the loss through LE/FD. A fair and balanced discussion should probably make that clear.
> > > > > >
> > > > > > Thank you very much for bringing up these great points! We will make sure to add the discussion to the revision of our final paper.
> > > > > >
> > > > > > While there is not a precise apples-to-apples comparison due to differences in the problem setup, in the following, we summarize our current understanding of how these approaches relate to and differ from ours and speculate about future work.
> > > > > >
> > > > > > [1] provides an alternative choice of regularization using LE and FD, it would be interesting to test in combination with our work in the future and see how the method works on a high-dimensional and noisy chaotic system when LE and FD are hard to compute reliably ([5]).
> > > > > >
> > > > > > [2] brings a unique insight into using a Fokker-Planck PDE for modeling the probability state for selected dynamical variables. However, since it requires estimating a high-dimensional probability distribution using finite mesh grids, it is challenging to scale the method to our high-dimensional problem.
> > > > > >
> > > > > > The architecture (RNNs) and teacher forcing (TF) [3,4] focused on could not be applied to our multi-environment setup without *major* modifications. Specifically, they focus on single environment dynamics and prove the optimal time interval of TF should depend on the LLE of the dynamics (eq. 17 in [3]). However, each trajectory in our training and testing dataset has its own LLE since we are working on multi-environment dynamics, which implies the direct application of [3,4] would result in a suboptimal choice of the time interval of TF and might impact the performance. In addition, we speculate that the high-noise setup we focus on might lead to further differences in performance.
> > > > > >
> > > > > > We thank you again for bringing up these valuable references, and we are especially delighted to study these contemporary works and see the increasing attention in the field of learning the chaotic attractors!
> > > > > > All these approaches are interesting approaches that lend insight into learning chaotic attractors using different techniques, all with relative advantages and disadvantages. Exploring the relative performance of these methods across settings would be a major experimental undertaking and beyond the scope of this submission, but it would be a fascinating study for future work and help us develop a more comprehensive understanding of different approaches.
> > > > > >
> > > > > > **References**
> > > > > >
> > > > > > [1] Constraining Chaos. Platt et al, April 2023.
> > > > > >
> > > > > > [2] Learning dynamics on invariant measures using PDE-constrained optimization. Botvinick-Greenhouse et al, Chaos, June 2023.
> > > > > >
> > > > > > [3] On the difficulty of learning chaotic dynamics with RNNs. Mikhaeil et al, NeurIPS 2022.
> > > > > >
> > > > > > [4] Generalized Teacher Forcing for Learning Chaotic Dynamics. Hess et al, June 2023.
> > > > > >
> > > > > > [5] Impracticality of a box-counting algorithm for calculating the dimensionality of strange attractors. Greenside, H. S. et al, Phys. Rev. A, 1982.

---

> > > > > > > ### Comment · Reviewer_GF1X · 2023-08-21
> > > > > > > **Thanks**
> > > > > > >
> > > > > > > Thank you. I understood the authors are considering a multi-environment setting, I’m not so sure though it makes such a big difference for the argument I was trying to make. Either way, I didn’t mean to imply by my remarks the authors necessarily benchmark against these other methods (although the high-noise setting would have been interesting given that some of the other methods excel even on noisy empirical data). I just wanted to point out that there are apparently different ways to achieve excellent long-term behavior on chaotic systems, even if one sticks to rMSE, and a balanced discussion and literature context should probably reflect that. This doesn’t take away from the authors’ own approach.
> > > > > > > I raised my score by 2.

---

> > > > > > > > ### Author Response · Authors · 2023-08-21
> > > > > > > > **Thank you for your reply and the fruitful discussion!**
> > > > > > > >
> > > > > > > > Dear reviewer, we are sincerely thankful for your constructive feedback, thoughtful comments, and appreciation of our work! Our fruitful discussion sheds light on the future work for learning the invariant attractors for chaotic dynamics. We truly value your feedback and will make sure to include the results of Lyapunov Spectrum and FD, as well as our valuable discussion of the related contemporary works, in our final paper. Thank you!

---

### Author Rebuttal · Authors · 2023-08-10

We would like to first thank the reviewers and ACs for their helpful comments and questions. We are happy to see that the reviewers generally appreciate the importance of the problem of training better emulators for chaotic dynamics and find the proposed methods and ideas well-motivated. In our response, we provide new experiments, evaluation metrics, and additional discussion to help clarify the properties of our two proposed approaches.

# Comparison of optimal transport and contrastive learning approaches

Several reviewers asked for a more detailed comparison between the two proposed approaches: physics-informed optimal transport (OT) and contrastive learning (CL). **Ultimately, both methods have strong performance according to a variety of metrics, but the CL approach requires no prior physical knowledge and is faster. Both approaches have significant advantages over the classical approach of minimizing rMSE, which fails to preserve important statistical characteristics of the system.**

We present both approaches here because they are both methods for encouraging the emulator to capture the long-term statistical behavior of the chaotic system. In fact, both approaches work by matching statistics computed from the emulator to statistics from the data. The primary conceptual difference between them is that the optimal transport approach takes advantage of prior domain knowledge about the setting to choose informative summary statistics, while the contrastive learning approach learns informative invariant statistics from the data alone.

## Physics-informed OT

### Advantages:
* If prior domain knowledge is available, OT can make good use of it and often performs better than CL when given highly informative summary statistics.
* OT does not require pre-training to learn a distance measure.
### Disadvantages:
* Without domain knowledge, an arbitrary uninformative summary statistic will often have no performance benefit.
* Because OT requires estimating the Wasserstein distance ($O(n^2\log(n))$ operations) for each training step, it is slower than the CL approach during emulator training. However, the scaling behavior is still reasonable, and the computation can be made even faster via sub-sampling.
## Unsupervised CL

### Advantages:
* CL does not require any prior domain knowledge and instead learns informative statistics directly from the data.
* CL is faster than OT during emulator training since it uses a pre-trained encoder for invariant statistics rather than computing distributional distances.
### Disadvantages:
* As an unsupervised representation learning method, CL requires a pre-training step to learn the informative statistics, which are later used for emulator training. *However, this fixed one-time cost is easily amortized.*
* In principle, CL relies on the data diversity of the multi-environment setting, although our new experiments show that we can still obtain good results even with a very minimal diversity of environments. *Also, the multi-environment setting is common in many scientific and engineering applications due to environmental noise or tunable control parameters, so this data diversity is often “free”.*
We have an existing discussion comparing the two approaches in the submitted supplement, and we will include a more detailed discussion in the final paper.

# New experiments and evaluation metrics

We have performed several new experiments that act as additional points of comparison and help us better understand the behavior of our methods under a variety of conditions:

1. For our OT approach (which uses the Sinkhorn loss), we test a reduced set of summary statistics, which shows how the quality of the summary statistic affects the performance of the method (Table R2). With an informative summary statistic, we find even a reduced set can still be helpful but, for a non-informative statistic, the OT method fails as expected.
2. For our CL approach, we test a multi-environment setting with reduced data diversity and find that the contrastive method still performs well under the reduced conditions (Table R3), which demonstrates robustness.
3. We also implement a variant of our OT approach that uses Maximum Mean Discrepancy (MMD) as a distributional distance rather than the Wasserstein distance. Using the same set of summary statistics, we find that MMD does not perform as well as the Wasserstein distance for training emulators (Table R4).

As suggested by the reviewers, we also add new evaluation metrics (Table R1) in the form of:

1. the leading Lyapunov exponent (LLE)—a dynamical quantity that measures how quickly the chaotic system becomes unpredictable; and
2. the fractal dimension—a characterization of the dimension of the attractor.

For the LLE, we report the relative absolute error $|\hat\lambda - \lambda|/|\lambda|$ between the model and the ground truth averaged over the test set. For the fractal dimension, we report the absolute error $|\hat D - D|$ between the estimated fractal dimension of the model and the ground truth averaged over the test set.

We generally find in high-noise settings that both of our approaches give lower LLE errors than an emulator trained on rMSE loss alone, and the contrastive approach has the lowest LLE error. This is despite the fact that our method is only encouraged to match invariant statistics of the final attractor rather than dynamical quantities. We also see evidence that the fractal dimension of the CL approach is closer to the true attractor. However, we note that the fractal dimension is difficult to reliably estimate for high-dimensional chaotic attractors [1].

The new experiments provide useful insights for future applications of our approaches, and the new evaluation metrics shed additional light on the features of chaos preserved by our trained emulators.

[1] Impracticality of a box-counting algorithm for calculating the dimensionality of strange attractors. Greenside, H. S. et al, Phys. Rev. A (1982).

---

> ### Author Response · Authors · 2023-08-16
> **Thank you for your review!**
>
> Dear reviewer, we truly value your feedback and are genuinely open to further discussions to ensure clarity of your question and concerns. And if you have any questions or need further clarifications, we would be very happy to assist and answer them! Thank you for your understanding.

---

> > ### Author Response · Authors · 2023-08-21
> > **Thank you for the discussion!**
> >
> > Dear reviewers and ACs, we are very grateful for your time and efforts in reviewing our paper and providing us with a fruitful discussion!
> >
> > We are excited to see that our reviewers appreciated our novel contributions to training better emulators for chaotic dynamics. Our results have been strengthened by the additional evaluation metrics and experiments suggested by several reviewers. We also had a very fruitful discussion that helped us clarify the motivations and the advantages of our approaches. We are also excited to see the growing interest in chaotic dynamics from contemporary works that will form the basis for future work in this area. We will make sure to include these additional results and discussions in our final paper.

---

### Decision · Program_Chairs · 2023-09-21

**Decision:**

Accept (poster)

**Comment:**

Although some reviewers have some technical concerns such as the weekness of the empirical evaluations in the first review, the authors have provided appropriate feedback in the discussion phase and addressed these concerns adequately. Finally, all reviewers agree that the paper has sufficient merit in the discussed problem. Therefore, I recommend acceptance of this paper for this submission.